# Gauss Law, Minimal Coupling and Fermionic PEPS for Lattice Gauge Theories

P. Emonts[1],
E. Zohar[2],

**1** Max-Planck-Institut für Quantenoptik, Hans-Kopfermann-Straße 1, 85748 Garching, Germany
**2** Racah Institute of Physics, The Hebrew University of Jerusalem 91904, Givat Ram, Jerusalem, Israel

December 20, 2019

## Abstract

In these lecture notes, we review some recent works on Hamiltonian lattice gauge theories, that involve, in particular, tensor network methods. The results reviewed here are tailored together in a slightly different way from the one used in the contexts where they were first introduced. We look at the Gauss law from two different points of view: for the gauge field, it is a differential equation, while from the matter point of view, on the other hand, it is a simple, explicit algebraic equation. We will review and discuss what these two points of view allow and do not allow us to do, in terms of unitarily gauging a pure-matter theory and eliminating the matter from a gauge theory, and relate that to the construction of PEPS (Projected Entangled Pair States) for lattice gauge theories.

## Contents

# 1  Introduction

Gauge symmetries, that provide the standard model's description of interactions, are fascinating. They have many interesting properties, that make them an excellent playground for endless studies of non-perturbative physics. They offer puzzling phenomena to study, understand, and solve, due to their rich symmetry and non-perturbative nature; perhaps the most famous one is the confinement of quarks in Quantum Chromodynamics (QCD) and other non-Abelian gauge theories [1].

Asymptotic freedom [2] tells us that the coupling of QCD is small at high energies, allowing one to use perturbation theory and Feynman diagrams to describe and understand collider physics. However, the other, non-perturbative low energy physics side, is yet to be understood. A very successful approach to that has been through lattice gauge theories [1, 3], where spacetime is discretized in a way that allows one to regularize the theory in a gauge invariant manner, and to perform successful Monte-Carlo calculations of many important physical properties – such as the hadronic spectrum [4]. On the other hand, as the Monte-Carlo calculations are carried out in Euclidean spacetime, they do not allow, in general, to consider real-time evolution and encounter the fermionic sign problem [5] in many physically relevant scenarios required, for example, for the study of exotic phases of QCD [6].

Recently, much effort has been put into lattice gauge theories from the quantum information and computation point of view, suggesting new ways to tackle these problems. One is quantum simulation [7–9], which suggests to map the gauge theory degrees of freedom to those of a controllable quantum system that could be used as an analog experiment, or a digital quantum computer for gauge theories. The other involves the application of tensor network tools, such as MPS or PEPS [10, 11].

Lattice gauge theories, in both methods, are approached in the Hamiltonian formalism, first introduced by Kogut and Susskind [12]. Unlike in path integral methods used widely in particle physics in general, and in lattice gauge theories in particular, the Hamiltonian formulation uses states and operators in Hilbert spaces, that have a very special structure due to the gauge symmetry.

Constructing a quantum simulator or a tensor network state involves an interesting challenge: one has to reconstruct a physical model from elementary building blocks. In this process, one faces the most fundamental elements of a theory, which is decomposed to its smallest ingredients. This, in some sense, is what we would like to discuss in this lecture: it will mostly be about enforcing gauge invariance on quantum states – the construction of tensor network states for lattice gauge theories, through a discussion of the Gauss law.

The key ingredient of gauge theories is the local nature of the symmetry: *a gauge symmetry is a local symmetry*, which means that they involve many (local) conservation laws, or constraints. The gauge symmetry is formulated, in both classical and quantum gauge theories, by means of the Gauss law, an equation – or a set of equations, one per space point – that relates the gauge field and the matter fields. Its simplest form, of classical electrodynamics, is

$$\nabla \cdot \mathbf{E}\left(\mathbf{x}\right) = \rho\left(\mathbf{x}\right) \tag{1}$$

– the divergence of the electric field $\mathbf{E}$ at each space point $\mathbf{x}$ equals the density of matter charges there, $\rho$. It can be made more complicated when the gauge field is non-Abelian, become an eigenvalue equation for the so-called *physical states* after quantization, or a difference equation for lattice gauge theories, but the essence is the same in all those cases: the matter is the source of electric fields.

The Gauss law is a static equation, even though its ingredients are, in general, time dependent. It involves no temporal derivatives; as the generator of a symmetry, it commutes with the Hamiltonian. It formulates constants of motion, rather than an equation of motion. Although it can be obtained from the Euler-Lagrange equations for fields, it is an artifact of the continuous Lagrangian formalism for fields, that puts space and time on an equal footing. However, it cannot be obtained through the Hamilton equations, but rather appears as a constraint accompanied by a Lagrange multiplier once a Legendre transformation into the Hamiltonian formalism is carried out. In lattice formulations with continuous time, which do not put time and space on an equal footing, it cannot be obtained as an equation of motion either. As it is a constraint, it means we can try to solve it, and if we manage to do that, we plug it into the other equations of motion to reduce the number of degrees of freedom.

In this lecture, we will discuss the nature and possibilities of solving the Gauss law, in two contexts. First, we will look at it as a differential equation for the gauge field, and see that it tells us that gauging a free matter theory (minimal coupling) could not be done, in the most general setting using a local unitary transformation. On the other hand, we can break the system into pieces with simple Gauss laws (and we will explain what "simple" means here), modify globally invariant Hamiltonians or states by gauging locally and unitarily each piece alone,

and then tailor everything together; this will be done using the Trotter-Suzuki decomposition for Hamiltonians, and construction of PEPS - Projected Entangled Pair States. Second, we will discuss the possibility of solving it for the matter, and using that for the elimination of the matter degrees of freedom, or, in case they are fermionic – their replacement by spins.

The lecture reviews the works [13–20].

## 2 Gauging Hamiltonians and Quantum States

### 2.1 Minimal Coupling and its Compact Lattice Formulation

A conventional textbook approach to gauge theories is to consider first a non-interacting matter field theory, with some global symmetry: the reader is suggested to naively try to act on a globally invariant Lagrangian or Hamiltonian with local transformations instead of global ones, only to quickly find out that the kinetic (derivative) term is not invariant under them. Then, in order to obtain something with a local symmetry nevertheless, another degree of freedom – the gauge field – is introduced as a geometric connection, modifying the previously quadratic, non-interacting terms into interaction terms of the matter with the field, in a procedure known as *minimal coupling*. At this point the gauge field has no dynamics; this could be introduced by including additional terms in the Hamiltonian or the Lagrangian.

The most famous example, perhaps, is the derivation of the Quantum Electrodynamics (QED) Lagrangian from that of the free Dirac theory [21]; one begins with

$$\mathcal{L}_D = \overline{\psi}\left(x\right)\left(i\slashed{\partial} - m\right)\psi\left(x\right), \tag{2}$$

which is symmetric under the global transformation $\psi\left(x\right) \rightarrow \psi\left(x\right)e^{i\Lambda}$. The slash notation $\slashed{B}$ is defined as $\slashed{B} \equiv \gamma^{\mu}B_{\mu}$, where $\gamma$ are the Dirac gamma matrices, regularly used for describing spin physics in the Dirac theory [21, Chapter 3]. In order to make (2) symmetric under local transformations of the form $\psi\left(x\right) \rightarrow \psi\left(x\right)e^{i\Lambda(x)}$, one introduces the gauge field $A_{\mu}\left(x\right)$ and replaces the regular derivative $\partial_{\mu}$ by the covariant one $D_{\mu} = \partial_{\mu} - iA_{\mu}\left(x\right)$, to obtain instead

$$\widetilde{\mathcal{L}}_D = \overline{\psi}\left(x\right)\left(i\slashed{D} - m\right)\psi\left(x\right). \tag{3}$$

Finally, the dynamics are obtained by adding the Maxwell part $\mathcal{L}_{EM} = -\frac{1}{4}F_{\mu\nu}\left(x\right)F^{\mu\nu}\left(x\right)$ (where $F_{\mu\nu}\left(x\right) = \partial_{\mu}A_{\nu}\left(x\right) - \partial_{\nu}A_{\mu}\left(x\right)$ is the gauge-invariant field strength tensor). The complete process could be summarized as

$$\mathcal{L}_D \longrightarrow \widetilde{\mathcal{L}}_D \longrightarrow \widetilde{\mathcal{L}}_D + \mathcal{L}_{EM}. \tag{4}$$

One could perform a similar procedure on the lattice. Consider, as a simple example, a two dimensional spatial lattice, with continuous time, as in the usual Hamiltonian formulation of lattice gauge theories [12]. We define a fermionic mode (creation operator) $\psi^{\dagger}\left(\mathbf{x}\right)$ at each vertex (lattice site) $\mathbf{x} \in \mathbb{Z}^2$. We introduce a staggered fermionic Hamiltonian, where one sublattice corresponds to particles and another to anti-particles; if one tunes the hopping coefficients in a slightly different way, which was not done here for the sake of simplicity, a (doubled) Dirac model is obtained in the continuum limit [22]:

$$H_f = M\sum_{\mathbf{x}}\left(-1\right)^{\mathbf{x}}\psi^{\dagger}\left(\mathbf{x}\right)\psi\left(\mathbf{x}\right) + \epsilon\sum_{\mathbf{x},i=1,2}\left(\psi^{\dagger}\left(\mathbf{x}\right)\psi\left(\mathbf{x}+\hat{\mathbf{e}}_i\right) + h.c.\right), \tag{5}$$

where $\hat{\mathbf{e}}_i$ is a unit vector in the direction $i = 1, 2$. $H_f$ is invariant under global phase transformations, generated by the total fermionic number

$$\mathcal{Q} = \sum_{\mathbf{x}}\psi^{\dagger}\left(\mathbf{x}\right)\psi\left(\mathbf{x}\right), \tag{6}$$

which is a conserved charge.

To make the symmetry local, we introduce on each link $\ell = \left(\mathbf{x}, i\right)$ of the lattice a new Hilbert space for the gauge field. In the notation $\ell = \left(\mathbf{x}, i\right)$, $\mathbf{x}$ is the lattice position from which the link is emanates to direction $i$. In a two-dimensional square lattice, $i$ can take the values $1, 2$. Along with the Hilbert space, we introduce two conjugate operators: the compact, angular vector potential $\phi\left(\mathbf{x}, i\right)$, and the electric field $E\left(\mathbf{x}, i\right)$ [3],

$$\left[\phi\left(\mathbf{x}, i\right), E\left(\mathbf{y}, j\right)\right] = i\delta_{\mathbf{x},\mathbf{y}}\delta_{ij}. \tag{7}$$

$E(\mathbf{x}, i)$ is a $U(1)$ angular momentum operator, conjugate to the angle $\phi(\mathbf{x}, i)$; we recognize the Hilbert space on each link of such a compact $U(1)$ lattice gauge theory as this of a particle moving on a ring.

As phase operators are not well defined, we work instead with the *group element operator*

$$U(\mathbf{x}, i) = e^{i\phi(\mathbf{x}, i)} \tag{8}$$

– playing the role of a Wilson line along the link.

On each link, the Hilbert space may be spanned, in particular, by two complete sets of states [23]: either the *magnetic* one, of group elements (or more accurately, in this case, parameters), $|\phi\rangle$, with the orthogonality relation

$$\langle \phi' | \phi \rangle = \delta(\phi' - \phi) \tag{9}$$

or the orthonormal basis of electric field eigenvalues $|\ell\rangle$ (representation basis),

$$E |\ell\rangle = \ell |\ell\rangle, \tag{10}$$

for every $\ell \in \mathbb{Z}$: the electric field is a non-bounded operator with an integer spectrum, and the group element operator serves as a unitary raising operator of it:

$$U |\ell\rangle = |\ell + 1\rangle. \tag{11}$$

In group theory terms, $\ell$ is simply an irreducible representation of $U(1)$,

$$\langle \phi | \ell \rangle = \frac{1}{\sqrt{2\pi}} e^{i\ell\phi} \tag{12}$$

- an eigenfunction of the Hamiltonian of a free particle on a ring, $H_{\text{ring}} \propto E^2$. $U = \int d\phi e^{i\phi} |\phi\rangle \langle \phi|$ is a fundamental representation group element operator (and similarly, one may define for any representation, $U^k = \int d\phi e^{ik\phi} |\phi\rangle \langle \phi|$, for which $U^k |\ell\rangle = |\ell + k\rangle$).

Having introduced the gauge field and its Hilbert space, we can minimally couple it to the matter, by modifying the Hamiltonian to

$$\widetilde{H}_f = M \sum_{\mathbf{x}} (-1)^{\mathbf{x}} \psi^\dagger(\mathbf{x}) \psi(\mathbf{x}) + \epsilon \sum_{\mathbf{x}, i=1,2} \left( \psi^\dagger(\mathbf{x}) U(\mathbf{x}, i) \psi(\mathbf{x} + \hat{\mathbf{e}}_i) + h.c. \right), \tag{13}$$

(where $(-1)^{\mathbf{x}} = (-1)^{x_1 + x_2}$) which is gauge invariant, i.e. invariant under local transformations generated by the *Gauss law operators*

$$\begin{aligned} \mathcal{G}(\mathbf{x}) &\equiv G(\mathbf{x}) - Q(\mathbf{x}) \\ &= \underbrace{E(\mathbf{x}, 1) + E(\mathbf{x}, 2) - E(\mathbf{x} - \hat{\mathbf{e}}_1, 1) - E(\mathbf{x} - \hat{\mathbf{e}}_2, 2)}_{G(\mathbf{x})} - Q(\mathbf{x}) \end{aligned} \tag{14}$$

where $G(\mathbf{x})$ is the lattice operator version of the divergence of the electric field at the vertex $\mathbf{x}$, and the staggered charge is

$$Q(\mathbf{x}) = \psi^\dagger(\mathbf{x}) \psi(\mathbf{x}) - \frac{1}{2} \left( 1 - (-1)^{\mathbf{x}} \right). \tag{15}$$

Finally, one can add dynamics to the gauge field, in the form of the Kogut Susskind (pure gauge) Hamiltonian $H_{KS} = H_E + H_B$ [3, 12], where

$$H_E = \frac{g^2}{2} \sum_{\mathbf{x}, i} E^2(\mathbf{x}, i), \tag{16}$$

$$H_B = -\frac{1}{g^2} \sum_{\mathbf{x}, i<j} \cos\left( \phi(\mathbf{x}, 1) + \phi(\mathbf{x} + \hat{\mathbf{e}}_1, 2) - \phi(\mathbf{x} + \hat{\mathbf{e}}_2, 1) - \phi(\mathbf{x}, 2) \right) \tag{17}$$

are the electric and magnetic Hamiltonians, respectively, with the coupling constant $g$. The complete process can be summarized by

$$H_f \longrightarrow \widetilde{H}_f \longrightarrow H = \widetilde{H}_f + H_{KS}. \tag{18}$$

## 2.2 Gauge Symmetry and the Physical Hilbert Space

The Hilbert space of a lattice gauge theory $\mathcal{H}$, as discussed above, could be decomposed as

$$\mathcal{H} \subset \mathcal{H}_{\mathrm{g}} \times \mathcal{H}_{\mathrm{f}}, \tag{19}$$

where $\mathcal{H}_{\mathrm{g}}$ is the Hilbert space of the gauge fields on the links, while $\mathcal{H}_{\mathrm{f}}$ represents the matter degrees of freedom on the vertices. In the fermionic case it is a Fock space, but one may consider other types of matter as well. However, not every state in the product space $\mathcal{H}_{\mathrm{g}} \times \mathcal{H}_{\mathrm{f}}$ is what we call "physical", due to the symmetry and its implications, which is why (19) is not an equality but rather an embedding. There have been different approaches to identify the gauge invariant sectors of the Hilbert space within the larger product space, e.g. by finding dualities between gauge invariant systems and spin systems [24]. More recently, such an embedding has been discussed in the context of entanglement entropy measurements [25]. In [26], a connection between the Hilbert space structure of gauge theories and tensor networks has been established, by using tensor networks for describing states in the gauge invariant part of the Hilbert space.

The gauge symmetry implies that

$$\left[\mathcal{G}\left(\mathbf{x}\right), H\right] = 0 \qquad \forall \mathbf{x} \tag{20}$$

– every *physical state* $|\psi\rangle$ is an eigenstate of all the Gauss law operators $\mathcal{G}\left(\mathbf{x}\right)$, with eigenvalues $q\left(\mathbf{x}\right)$ which we call *static charges*:

$$\mathcal{G}\left(\mathbf{x}\right)|\psi\rangle = q\left(\mathbf{x}\right)|\psi\rangle \quad \Longleftrightarrow \quad G\left(\mathbf{x}\right)|\psi\rangle = \left(Q\left(\mathbf{x}\right) + q\left(\mathbf{x}\right)\right)|\psi\rangle. \tag{21}$$

This eigenvalue equation is simply the Gauss law – a quantum lattice version of the well-known continuum classical equation (1). The static charge configurations $\left\{q\left(\mathbf{x}\right)\right\}$ are constants of motion, splitting the physical Hilbert space into different superselection sectors,

$$\mathcal{H} = \bigcup_{\left\{q\left(\mathbf{x}\right)\right\}} \mathcal{H}\left(\left\{q\left(\mathbf{x}\right)\right\}\right) \tag{22}$$

that are not mixed by the dynamics. When one acts with a local gauge transformation on a state, a global phase depending on the static charge configuration sector $\mathcal{H}\left(\left\{q\left(\mathbf{x}\right)\right\}\right)$ to which it belongs will appear. Due to the superselection rule we do not discuss superpositions, and thus only global phases can appear, and they play no role in quantum mechanics.

Therefore, in some sense, all the states in $\mathcal{H}$ are gauge invariant – but with different global phases, depending on the static charges. In our context, however, a *gauge invariant state* will be such that is strictly invariant, i.e. without a global phase. This generalizes also to other, non-Abelian groups, where static charges imply multiplets of states that are connected through gauge transformation, unless the static charges are group singlets. From now on, when we discuss the *physical* or the *gauge invariant Hilbert space*, we will refer only to the $\mathcal{H}\left(\left\{q\left(\mathbf{x}\right) = 0\right\}\right)$ sector.

A general physical state $|\psi\rangle \in \mathcal{H}\left(\left\{q\left(\mathbf{x}\right) = 0\right\}\right)$ satisfies

$$G\left(\mathbf{x}\right)|\psi\rangle = Q\left(\mathbf{x}\right)|\psi\rangle \tag{23}$$

and thus it may be expressed as a superposition of product states of gauge field and matter, with the same eigenvalues for $G\left(\mathbf{x}\right), Q\left(\mathbf{x}\right)$:

$$|\psi\rangle = \sum_{\left\{Q\left(\mathbf{x}\right)\right\}} \alpha\left(\left\{Q\left(\mathbf{x}\right)\right\}\right)\left|E\left(\left\{Q\left(\mathbf{x}\right)\right\}\right)\right\rangle_{\mathrm{Gauge}} \otimes \left|\left\{Q\left(\mathbf{x}\right)\right\}\right\rangle_{\mathrm{Matter}}, \tag{24}$$

where $\alpha\left(\left\{Q\left(\mathbf{x}\right)\right\}\right) \in \mathbb{C}$ are the weights for different states in the superposition. However, note that once $\left|\left\{Q\left(\mathbf{x}\right)\right\}\right\rangle_{\mathrm{Matter}}$ is fixed, there is more than one choice for $\left|E\left(\left\{Q\left(\mathbf{x}\right)\right\}\right)\right\rangle_{\mathrm{Gauge}}$ (it is, in general, a superposition).

Which means that a unitary gauging map of the form

$$
\left( \sum_{\{Q(\mathbf{x})\}} \alpha\left(\{Q(\mathbf{x})\}\right) \left|\{Q(\mathbf{x})\}\right\rangle_{\text{Matter}} \right) \otimes |0\rangle_{\text{Gauge}}
$$
$$
\longrightarrow \sum_{\{Q(\mathbf{x})\}} \alpha\left(\{Q(\mathbf{x})\}\right) \left|E\left(\{Q(\mathbf{x})\}\right)\right\rangle_{\text{Gauge}} \left|\{Q(\mathbf{x})\}\right\rangle_{\text{Matter}}
\tag{25}
$$

(minimal coupling) is, in general, not possible. On the other hand, once $\left|E\left(\{Q(\mathbf{x})\}\right)\right\rangle_{\text{Gauge}}$ is fixed, there could

be a unique choice for $\left|\{Q(\mathbf{x})\}\right\rangle_{\text{Matter}}$ (depending on the particular model; this is true for our $U(1)$ staggered case, but not in general). Therefore a matter eliminating transformation,

$$
\sum_{\{Q(\mathbf{x})\}} \alpha\left(\{Q(\mathbf{x})\}\right) \left|E\left(\{Q(\mathbf{x})\}\right)\right\rangle_{\text{Gauge}} \left|\{Q(\mathbf{x})\}\right\rangle_{\text{Matter}}
$$
$$
\longrightarrow |0\rangle_{\text{Matter}} \otimes \sum_{\{Q(\mathbf{x})\}} \alpha\left(\{Q(\mathbf{x})\}\right) \left|E\left(\{Q(\mathbf{x})\}\right)\right\rangle_{\text{Gauge}}
\tag{26}
$$

exists.

This is strongly related to the Gauss law and its properties – as we shall discuss below.

## 2.3 Can Minimal Coupling be Unitary and Local?

In quantum mechanics we like unitary transformations. In particular, when two Hamiltonians are connected by a unitary transformation (just like any other pair of observables), they will have the same spectrum.

Let us focus on the first step, $H_f \longrightarrow \widetilde{H}_f$. It is clear that $\left[\widetilde{H}_f, U(\mathbf{x}, i)\right] = 0$ as no gauge field dynamics is introduced at this point. Could we then find a unitary gauging, or minimal coupling transformation $\mathcal{U}_G$ – for which $\mathcal{U}_G H_f \mathcal{U}_G^\dagger = \widetilde{H}_f$? To make the situation even simpler, let us first only deal with a single link, with two fermionic modes $\psi, \chi$ on its edges. The globally invariant Hamiltonian is

$$
H_2 = M\left(\psi^\dagger\psi - \chi^\dagger\chi\right) + \epsilon\left(\psi^\dagger\chi + \chi^\dagger\psi\right)
\tag{27}
$$

with a global symmetry generated by

$$
Q_2 = \psi^\dagger\psi + \chi^\dagger\chi.
\tag{28}
$$

Gauging it will lead to

$$
\begin{aligned}
\widetilde{H}_2 &= M\left(\psi^\dagger\psi - \chi^\dagger\chi\right) + \epsilon\left(\psi^\dagger U\chi + \chi^\dagger U^\dagger\psi\right) \\
&= M\left(\psi^\dagger\psi - \chi^\dagger\chi\right) + \epsilon\left(\psi^\dagger e^{i\phi}\chi + \chi^\dagger e^{-i\phi}\psi\right)
\end{aligned}
\tag{29}
$$

with the symmetry charges (Gauss law operators)

$$
\mathcal{G}_\psi = E - \psi^\dagger\psi \quad ; \quad \mathcal{G}_\chi = -E - \left(\chi^\dagger\chi - 1\right)
\tag{30}
$$

Our initial Hilbert space is $\mathcal{H}_f$, a subspace of the fermionic Hilbert space, that contains the globally invariant fermionic states. Gauging maps it to $\mathcal{H}_{\text{phys}}$, a subspace of the product space $\mathcal{H}_f \times \mathcal{H}_g$ (where $\mathcal{H}_g$ is the link's Hilbert space), which is invariant under the transformations generated by $\mathcal{G}_\psi$ and $\mathcal{G}_\chi$. We will replace the initial space by $\mathcal{H}_f \times \{|0\rangle\}$, where $E|0\rangle = 0$. We would like to construct a unitary transformation $\mathcal{U}_G$, that maps $\mathcal{H}_f \times \{|0\rangle\}$ to $\mathcal{H}_{\text{phys}}$ – that is, it will entangle the vertices and link degrees of freedom, in a way that respects the symmetry generated by $\mathcal{G}_\psi$ and $\mathcal{G}_\chi$.

We will have to act on the state $|0\rangle$ in a way that will introduce the desired conservation laws. Since they have to be consistent with each other, we need some relation between the two fermionic number operators, but this is satisfied by our initial, globally invariant elements of $\mathcal{H}_f$. So we simply have to identify $E$ with $\psi^\dagger\psi$ or $\chi^\dagger\chi - 1$: that is, to raise the electric field on the link by an amount $\psi^\dagger\psi$, or lower it by $\chi^\dagger\chi - 1$. Let us choose the first option. This is done by the unitary [18]

$$\mathcal{U}_G = \int d\phi e^{i\phi\psi^\dagger\psi} |\phi\rangle\langle\phi|,\tag{31}$$

where $|\phi\rangle$ is a phase eigenstate, such that

$$U = \int d\phi e^{i\phi} |\phi\rangle\langle\phi|.\tag{32}$$

It is a controlled operation that could be seen either as raising the electric field $E$ with respect to the fermionic charge $\psi^\dagger\psi$, or as rotating the fermionic operators $\psi^\dagger,\psi$ with respect to the phase operator $\phi$. While the first interpretation suits the point of view of the quantum state, the second one suits that of the Hamiltonian, explaining us intuitively that, indeed,

$$\mathcal{U}_G H_2 \mathcal{U}_G^\dagger = \mathcal{U}_G \left(H_2 \otimes \mathbf{1}_{\text{link}}\right)\mathcal{U}_G^\dagger = \widetilde{H}_2\tag{33}$$

– so we see, that unitary gauging was obtained by solving the simple Gauss law (30) which was both local and uniquely solvable.

Could we now extend this to the Hamiltonian $H_f$ in arbitrary dimensions? The answer is no. Of course, for each link $\ell = (\mathbf{x}, i)$ we can define a gauging transformation

$$\mathcal{U}_G(\ell) = \mathcal{U}_G(\mathbf{x}, i) = \int d\phi e^{i\phi\psi^\dagger(\mathbf{x})\psi(\mathbf{x})} |\phi\rangle\langle\phi|_\ell\tag{34}$$

such that

$$\mathcal{U}_G(\mathbf{x}, i)\psi^\dagger(\mathbf{x})\psi(\mathbf{x}+\hat{\mathbf{e}}_i)\mathcal{U}_G^\dagger(\mathbf{x}, i) = \psi^\dagger(\mathbf{x})U(\mathbf{x}, i)\psi(\mathbf{x}+\hat{\mathbf{e}}_i).\tag{35}$$

However, each mode appears in more than one link, and applying the product $\prod_\ell \mathcal{U}_G(\ell)$ to the whole Hamiltonian $H_f$ will give rise to a very complicated and messy expression which is not $\widetilde{H}_f$: since now each Gauss law involves more than one electric field, all fermionic operators will be rotated with respect to the gauge fields on all the links around them.

Let us move on more slowly then, and extend our one link system to an open line. The globally invariant Hamiltonian is

$$H_{1d} = M\sum_x (-1)^x \psi^\dagger(x)\psi(x) + \epsilon\left(\psi^\dagger(x)\psi(x+1) + h.c.\right)\tag{36}$$

with a global symmetry generated by

$$Q_{1d} = \sum_x \psi^\dagger(x)\psi(x)\tag{37}$$

Gauging it will lead to

$$\widetilde{H}_{1d} = M\sum_x (-1)^x \psi^\dagger(x)\psi(x) + \epsilon\left(\psi^\dagger(x)U(x)\psi(x+1) + h.c.\right)\tag{38}$$

$$= M\sum_x (-1)^x \psi^\dagger(x)\psi(x) + \epsilon\left(\psi^\dagger(x)e^{i\phi(x)}\psi(x+1) + h.c.\right)\tag{39}$$

with the symmetry charges (Gauss law operators)

$$\mathcal{G}(x) = \underbrace{E(x) - E(x-1)}_{G(x)} - Q(x)\tag{40}$$

We can solve for the electric field in a non-local way, e.g. by

$$E(x) = \sum_{y<x} Q(y).\tag{41}$$

If we like to repeat the single link procedure, we will need to put initially a trivial $|0\rangle$ state on each link, and then change it by an amount of $\sum\limits_{y<x} Q(y)$. This leads to a nonlocal gauging transformation, of the form

$$\mathcal{U}_G^{(1)} = \int \left( \prod_x d\phi_x \right) e^{-i\sum_x \left( \sum_{y=0}^{x-1} \phi(y) \right) \psi^\dagger(x)\psi(x)} |\{\phi\}\rangle \langle\{\phi\}| \tag{42}$$

which gives us what we want. Its inverse form can be used for eliminating the gauge field degrees of freedom in the one dimensional case, when one starts with the gauge invariant theory, but it will introduce some nonlocal interactions (from $H_E$). This was used both for MPS computations [27] and quantum simulation [28].

We have seen that in the case of a single link we can gauge with a unitary – a controlled operation – because the Gauss law has a unique solution. For a one dimensional system, we can gauge by a unitary transformation again, since a unique solution exists, but we lose locality. In more dimensions we cannot do even that anymore.

In order to understand it better, and in a more general way, let us look again at the equation that is responsible for all that – the Gauss law. Either its lattice form (14), or the continuum form,

$$\nabla \cdot \mathbf{E}(\mathbf{x}) = \rho(\mathbf{x}). \tag{43}$$

This is the equation that manifests the local symmetry, introduced to the system in the minimal coupling procedure, relating the gauge (electric) field to the matter, that existed before. In the process of gauging, we "complete" a physical setting that only had the matter field to one that involves the gauge field too.

Suppose, we wish to gauge a globally invariant Hamiltonian, or a state, by introducing a gauge field that will satisfy (43). For that, we have to solve the equation for the electric field. However, such a solution is, in general, not unique: (43) is a single, non-vector differential (or difference, on the lattice) equation, for a vector field. A general gauge invariant state could be a superposition of many different states satisfying the Gauss law, even if the matter charges are fixed. If we have no unique way to determine the gauge field configuration from the matter, a unitary transformation cannot be used for gauging.

We see, therefore, that gauging by a local unitary will generally not work: Minimally coupling is generally not a unitary modification of the system. The only scenarios that allow one to gauge using a unitary and local transformation are those in which the Hamiltonian or the state to be gauged are constructed out of separate, decoupled local parts, each of which could be separately gauged in a unique way. Only after gauging, one tailors the already gauged pieces together, and introduces nonlocal correlations. That is, we do not transform the whole object (Hamiltonian / state), but rather its building blocks, before connecting them.

## 2.4 The Case of Hamiltonians: Trotterized Gauge Invariant Dynamics

From a Hamiltonian point of view, unitary gauging of separate building blocks can be achieved using trotterized time evolution. The Trotter-Suzuki decomposition [29], which is very useful for quantum simulation [30], allows one to approximate the time evolution of a Hamiltonian by a product of Trotter steps: short time evolutions of different parts of the Hamiltonian, that do not necessarily commute. It is exact if each Trotter step is infinitesimally short; otherwise, for finitely short steps, error bounds can be computed.

For our purposes, we will break the Hamiltonians into separate link contributions, $\widetilde{H}_f = \sum\limits_\ell \widetilde{H}_{f,\ell}$ and $H_f = \sum\limits_\ell H_{f,\ell}$. This decomposition allows us to gauge separately:

$$\mathcal{U}_G(\ell) H_{f,\ell} \mathcal{U}_G^\dagger(\ell) = \widetilde{H}_{f,\ell} \tag{44}$$

such that

$$e^{-i\widetilde{H}_f t} = \lim_{N\to\infty} \left[ \prod_\ell e^{-i\widetilde{H}_\ell t/N} \right]^N = \lim_{N\to\infty} \left[ \prod_\ell \left( \mathcal{U}_G(\ell) e^{-iH_\ell t/N} \mathcal{U}_G^\dagger(\ell) \right) \right]^N. \tag{45}$$

Choosing a finite but large $N$ this can be used as an approximate time evolution, as widely done for quantum

simulation purposes in other contexts,

$$e^{-i\widetilde{H}_f t} \approx \left[ \prod_\ell \left( \mathcal{U}_G(\ell) e^{-iH_\ell t/N} \mathcal{U}_G^\dagger(\ell) \right) \right]^N. \tag{46}$$

Why does that work? Because every Trotter step involves the dynamics of a single link Hamiltonian, just like $H_2$ we considered above (27), and its gauging gives rise to simple, one dimensional Gauss laws generated by operators such as those of (30) rather than the ambiguous, vector Gauss laws of the general case (14). They are only "switched on" once a sequence of Trotter steps is considered, as symmetries of the complete time evolution.

Some errors due to the trotterization (non-commutativity of intersecting links contributions to the Hamiltonian) will arise; note, however, that these errors have nothing to do with the gauge symmetry – each evolution step is already gauge invariant and thus, although we obtain an approximate time evolution, the symmetry is exact. This was used in [16, 17, 31] for digital quantum simulation of lattice gauge theories (using a more "economical" ways of breaking the Hamiltonian to less pieces; the important thing is that intersecting links, involving common fermions, will not be included in the same Trotter step).

## 2.5 The Case of Quantum States: Gauging PEPS

We can also consider the states' point of view: one can try to construct the state from some local ingredients, that enable separate, individual actions of gauging transformations, and then, after gauging, project all the local pieces together in a way that forms a non-trivial, interacting state.

This can be done in the process we shall describe now. As we shall see, the states we construct in this way are nothing but PEPS – Projected Entangled Pair States [10] – but with a local symmetry, following the procedure of [14].

*Step 1.* At each site, construct a state $\big|A(\mathbf{x})\big\rangle$ that will involve the physical matter degree of freedom, and some virtual degrees of freedom as well. The virtual degrees of freedom are associated with the edges of links starting or ending at the vertex, and out of them we will construct the *virtual electric fields*, $E_0(z)$, where $z = r, u, l, d$ denotes the edge: right, up (outgoing) and left, down (ingoing). The state will be constructed in a way that a Gauss law will be satisfied by the physical charge and the virtual electric fields: i.e., we demand that it will be invariant under transformations generated by

$$\mathcal{G}_0 \equiv G_0 - Q = E_0(r) + E_0(u) - E_0(l) - E_0(d) - Q, \tag{47}$$

where $Q$ is the charge associated with the physical matter degree of freedom at the vertex, which does not have to be fermionic. At each vertex we can define the virtual transformations

$$W_z(\Lambda) = e^{i\Lambda E_0(z)} \tag{48}$$

and the physical one

$$V(\Lambda) = e^{i\Lambda Q} \tag{49}$$

such that

$$e^{i\Lambda \mathcal{G}_0} = W_r W_u W_l^\dagger W_d^\dagger V^\dagger. \tag{50}$$

This allows us to write the symmetry condition of $\big|A(\mathbf{x})\big\rangle$ in the conventional form used in the context of PEPS; at each vertex, the action with a group transformation on the physical degree of freedom is equivalent to acting on the virtual ones,

$$V\left|A\right\rangle = W_r W_u W_l^\dagger W_d^\dagger \left|A\right\rangle. \tag{51}$$

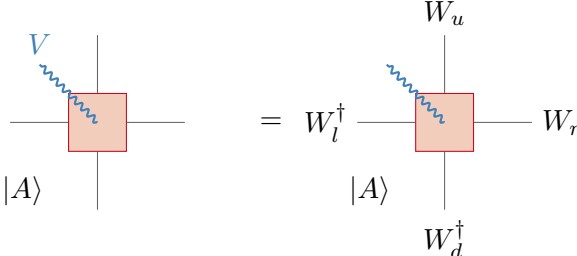

One nice and useful feature of tensor networks is their pictorial representation. All manipulations that are carried out with tensor networks can be conveniently depicted. Thus, the drawing below (51) yields the same information as the equation itself. Readers who are not familiar with this notation can find a short introduction in appendix A or in [11, 32]. In these notes, we chose some conventions for the legs to make the understanding of tensor network figures as easy as possible. Virtual legs are depicted as thin, gray lines. The physical legs related to fermions are drawn as thick, blue, wavy lines, while legs related to the local gauging are shown as thick, black, wavy lines. Projectors are colored in green and fiducial states, i.e. states that are located on the vertices, are shown in red.

*Step 2.* We take a product of the local matter states $\left|A\left(\mathbf{x}\right)\right\rangle$ with the gauge field states $\left|0\right\rangle$ on all the links starting at that vertex –

$$\left|A\left(\mathbf{x}\right)\right\rangle \longrightarrow \left|A\left(\mathbf{x}\right)\right\rangle \left|0\right\rangle_{\mathbf{x},1} \left|0\right\rangle_{\mathbf{x},2} \equiv \left|A\left(\mathbf{x}\right)\right\rangle \left|0\right\rangle_{\mathbf{x}} \tag{52}$$

*Step 3.* On each $\left|A\left(\mathbf{x}\right)\right\rangle$ we act with two gauging transformations $\mathcal{U}_G$, rotating (and entangling) the virtual degrees of freedom corresponding to links beginning at the vertex with respect to the gauge field degrees of freedom on that link: $\mathcal{U}_G\left(\mathbf{x},1\right)\mathcal{U}_G\left(\mathbf{x},2\right)$. The local states obtained by that,

$$\left|A_G\left(\mathbf{x}\right)\right\rangle = \mathcal{U}_G\left(\mathbf{x},1\right)\mathcal{U}_G\left(\mathbf{x},2\right)\left|A\left(\mathbf{x}\right)\right\rangle \left|0\right\rangle_{\mathbf{x}} \tag{53}$$

will obey two symmetries, connecting the physical fields and the virtual fields on the outgoing links,

$$V_{r,u}\left|A_G\right\rangle = W_{r,u}\left|A_G\right\rangle \tag{54}$$

where

$$V_r\left(\Lambda\right) = e^{i\Lambda E\left(\mathbf{x},1\right)}, \quad V_u\left(\Lambda\right) = e^{i\Lambda E\left(\mathbf{x},2\right)}. \tag{55}$$

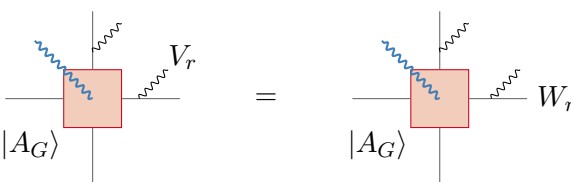

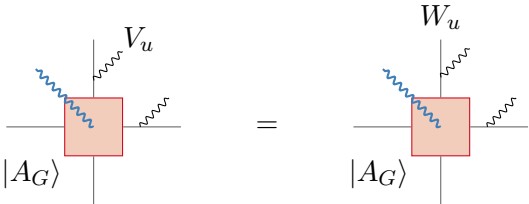

These are generated by operators such as those we considered in the one-link case (30), i.e. connecting each physical electric field only to one rotated degree of freedom (the virtual one in this case) and thus gauging is possible. As a result, these states will also obey modified Gauss laws, in which the physical electric fields on the outgoing links (up, right) replace the virtual ones, while the virtual fields remain for the ingoing links (down, left):

$$V\left|A_G\right\rangle = V_r V_u W_l^\dagger W_d^\dagger \left|A_G\right\rangle \tag{56}$$

These Gauss laws are already "half-way" to complete physical ones; we only need to contract the local states to one another, as done in the next step.

*Step 4.* We project the virtual ingredients into maximally entangled pairs connecting the edges of links $|\omega_\ell\rangle$, satisfying the symmetry relations

$$W_r^\dagger(\mathbf{x}) W_l(\mathbf{x} + \hat{\mathbf{e}}_1) |\omega_{\mathbf{x},1}\rangle = |\omega_{\mathbf{x},1}\rangle \tag{57}$$

$$W_u^\dagger(\mathbf{x}) W_d(\mathbf{x} + \hat{\mathbf{e}}_2) |\omega_{\mathbf{x},2}\rangle = |\omega_{\mathbf{x},2}\rangle \tag{58}$$

In the pictorial language of tensor networks, we denote the projector $\omega$ as a green square. The application of operators is marked by writing their names next to the legs.

This will remove all the virtual degrees of freedom, after having used them for contracting the whole state, which is not a trivial product state anymore.

The final state

$$|\psi\rangle = \underbrace{\otimes_\ell \langle\omega_\ell|}_{\text{Step 4}} \underbrace{\prod_\ell \mathcal{U}_G(\ell)}_{\text{Step 3}} \underbrace{\otimes_{\mathbf{x}} \underbrace{|A(\mathbf{x})\rangle}_{\text{Step 1}} |0\rangle_{\mathbf{x}}}_{\text{Step 2}} \tag{59}$$

will have the desired physical symmetry – local gauge invariance.

Steps 1 and 4 can be recognized as the construction of a PEPS: we project local states with some virtual symmetry to maximally entangled pair states. In fact, this is a PEPS as well – a gauged one. The lesson we learn from that is that in order to obtain a lattice gauge theory PEPS, we can first write down a globally invariant PEPS, describing the matter degrees of freedom of the desired gauge theory,

$$|\psi_0\rangle = \underbrace{\otimes_\ell \langle\omega_\ell|}_{\text{Step 4}} \underbrace{\otimes_{\mathbf{x}} |A(\mathbf{x})\rangle}_{\text{Step 1}} \tag{60}$$

out of which the gauge invariant PEPS, with dynamical gauge fields, may be obtained in the process described above, that "interrupts" the usual PEPS construction in the form of step 2 and step 3. This is exactly what we discussed above: gauging, or minimally coupling, by a unitary transformation, must be done before we contract the local ingredients, that is, on a product state that allows us to act separately on different links. Only after gauging we project to a nontrivial physical state of the whole lattice, at step 4: $|\psi\rangle$ is not gauged by directly acting with a unitary transformation on $|\psi_0\rangle$ ; rather, a modification is required before the system's ingredients are connected.

The nice "feature" that we get is that many symmetry properties of the original, globally invariant PEPS $|\psi_0\rangle$ "survive" the gauging procedure, such as spatial symmetries (translation, lattice rotation, lattice inversion etc. – anything that might have existed prior to gauging) with the right modification for the gauge field transformation laws (e.g. translation invariance of $|\psi_0\rangle$ can be made charge conjugation symmetry of $|\psi\rangle$). And, of course, the global symmetry becomes local. Therefore, it is enough to construct a state with a global symmetry, and the one with local gauge invariance is obtained immediately.

For the sake of completeness, and for readers not familiar with PEPS, let us show that the PEPS $|\psi_0\rangle$ and $|\psi\rangle$ indeed have the right symmetries. First, let us transform $|\psi_0\rangle$ with a global transformation and see what happens:

$$e^{i\Lambda\sum_{\mathbf{x}}Q(\mathbf{x})}\left|\psi_0\right\rangle = \underset{\ell}{\otimes}\left\langle\omega_\ell\right|\underset{\mathbf{x}}{\otimes}\left(V\left(\mathbf{x}\right)\left|A\left(\mathbf{x}\right)\right\rangle\right)$$

$$= \underset{\ell}{\otimes}\left\langle\omega_\ell\right|\underset{\mathbf{x}}{\otimes}\left(W_r\left(\mathbf{x}\right)W_u\left(\mathbf{x}\right)W_l^\dagger\left(\mathbf{x}\right)W_d^\dagger\left(\mathbf{x}\right)\left|A\left(\mathbf{x}\right)\right\rangle\right)$$

$$= \underset{\mathbf{x}}{\otimes}\left(\left\langle\omega_{\mathbf{x},1}\right|W_r\left(\mathbf{x}\right)W_l^\dagger\left(\mathbf{x}+\hat{\mathbf{e}}_1\right)\left\langle\omega_{\mathbf{x},2}\right|W_u\left(\mathbf{x}\right)W_d^\dagger\left(\mathbf{x}+\hat{\mathbf{e}}_2\right)\right)\underset{\mathbf{x}}{\otimes}\left|A\left(\mathbf{x}\right)\right\rangle \tag{61}$$

$$= \underset{\ell}{\otimes}\left|\omega_\ell\right\rangle\underset{\mathbf{x}}{\otimes}\left|A\left(\mathbf{x}\right)\right\rangle$$

$$= \left|\psi_0\right\rangle$$

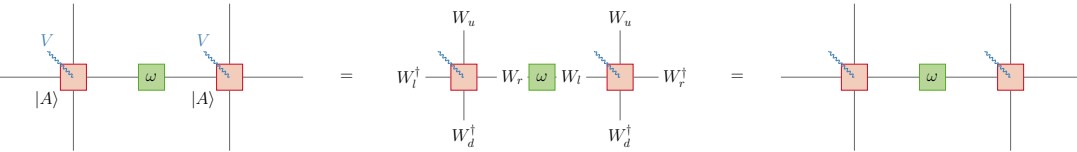

– the transition from the first line to the second is thanks to the symmetry property of $\left|A\left(\mathbf{x}\right)\right\rangle$; from the third to the fourth, it is the symmetry property of the bond states $\left\langle\omega_\ell\right|$.

In the local case,

$$e^{i\Lambda\mathcal{G}(\mathbf{x}_0)}\left|\psi\right\rangle = \underset{\ell}{\otimes}\left\langle\omega_\ell\right|V_r\left(\mathbf{x}_0\right)V_u\left(\mathbf{x}_0\right)V^\dagger\left(\mathbf{x}_0\right)\left|A_G\left(\mathbf{x}_0\right)\right\rangle$$

$$V_r^\dagger\left(\mathbf{x}_0-\hat{\mathbf{e}}_1\right)\left|A_G\left(\mathbf{x}_0-\hat{\mathbf{e}}_1\right)\right\rangle$$

$$V_u^\dagger\left(\mathbf{x}_0-\hat{\mathbf{e}}_2\right)\left|A_G\left(\mathbf{x}_0-\hat{\mathbf{e}}_2\right)\right\rangle\underset{\substack{\mathbf{x}\neq\mathbf{x}_0,\\\mathbf{x}_0-\hat{\mathbf{e}}_1,\\\mathbf{x}_0-\hat{\mathbf{e}}_2}}{\otimes}\left|A_G\left(\mathbf{x}\right)\right\rangle$$

$$= \underset{\ell}{\otimes}\left\langle\omega_\ell\right|W_l\left(\mathbf{x}_0\right)W_d\left(\mathbf{x}_0\right)\left|A_G\left(\mathbf{x}_0\right)\right\rangle$$

$$W_r^\dagger\left(\mathbf{x}_0-\hat{\mathbf{e}}_1\right)\left|A_G\left(\mathbf{x}_0-\hat{\mathbf{e}}_1\right)\right\rangle \tag{62}$$

$$W_u^\dagger\left(\mathbf{x}_0-\hat{\mathbf{e}}_2\right)\left|A_G\left(\mathbf{x}_0-\hat{\mathbf{e}}_2\right)\right\rangle\underset{\substack{\mathbf{x}\neq\mathbf{x}_0,\\\mathbf{x}_0-\hat{\mathbf{e}}_1,\\\mathbf{x}_0-\hat{\mathbf{e}}_2}}{\otimes}\left|A_G\left(\mathbf{x}\right)\right\rangle$$

$$= \underset{\ell}{\otimes}\left\langle\omega_\ell\right|W_l\left(\mathbf{x}_0\right)W_r^\dagger\left(\mathbf{x}_0-\hat{\mathbf{e}}_1\right)W_d\left(\mathbf{x}_0\right)W_u^\dagger\left(\mathbf{x}_0-\hat{\mathbf{e}}_2\right)\underset{\mathbf{x}}{\otimes}\left|A_G\left(\mathbf{x}\right)\right\rangle$$

$$= \underset{\ell}{\otimes}\left\langle\omega_\ell\right|\underset{\mathbf{x}}{\otimes}\left|A_G\left(\mathbf{x}\right)\right\rangle$$

$$= \left|\psi\right\rangle$$

As a final remark, note that this formulation can also be used for the construction of pure gauge states, with no physical matter: in that case, the states $\left|A\left(\mathbf{x}\right)\right\rangle$ will have no physical degrees of freedom, only virtual ones that enforce the Gauss law and connect the links.

### 2.5.1 Example

As an example [14], let us construct a very simple PEPS as follows. At each vertex, we use a physical Hilbert space spanned by states of the form $\left\{\left|p\right\rangle\right\}_{p=-J}^{J}$ (eigenstates of $Q$, labeled by its eigenvalues) and virtual ones with $\left\{\left|v\right\rangle\right\}_{v=-j}^{j}$ (similarly, eigenstates of the virtual electric fields $E_0$) where the integers $j, J$ can be either finite or

infinite. The symmetry is obtained for

$$|A\rangle = A^p_{ruld}\,|p\rangle\,|r,u,l,d\rangle\,,\tag{63}$$

where $|r,u,l,d\rangle$ is a product of four virtual basis states, $\{|v\rangle\}^j_{v=-j}$ , if

$$A^p_{ruld} \propto \delta\left(r+u,l+d+p\right)\tag{64}$$

– the Kronecker delta enforces the Gauss law (in general, the tensor elements $A^p_{ruld}$ should be proportional to the right combination of Clebsch-Gordan coefficients that correspond to the correct combination of representations [14]; in this simple $U(1)$ case, the Clebsch-Gordan coefficients are just Kronecker deltas). The gauging transformation will simply be

$$\mathcal{U}_G\left(\ell\right) = \begin{cases} \int d\phi\, e^{i\phi(\mathbf{x},1)E_0(r,\mathbf{x})}\,|\phi\rangle\,\langle\phi|_\ell & \ell\text{ horizontal} \\ \int d\phi\, e^{i\phi(\mathbf{x},2)E_0(u,\mathbf{x})}\,|\phi\rangle\,\langle\phi|_\ell & \ell\text{ vertical.} \end{cases}\tag{65}$$

Note that as a result of the gauging that identifies the physical electric fields with the virtual ones, $|E|\leq j$, and hence *if we have a finite truncation for the virtual electric field, the physical electric field will be equally truncated as well.*

Finally, the maximally entangled pair states on which we project will be

$$|\omega_\ell\rangle = \begin{cases} \sum\limits_{|v|\leq j}|v\rangle_{\mathbf{x},r}\,|v\rangle_{\mathbf{x}+\hat{\mathbf{e}}_1,l} & \ell\text{ horizontal} \\ \sum\limits_{|v|\leq j}|v\rangle_{\mathbf{x},u}\,|v\rangle_{\mathbf{x}+\hat{\mathbf{e}}_2,d} & \ell\text{ vertical.} \end{cases}\tag{66}$$

# 3    Construction of Gauge Invariant Fermionic PEPS (fPEPS)

While the above procedure for gauging a PEPS works for any PEPS with a global symmetry as $|\psi_0\rangle$, we shall focus now on a more particular case, of gauged Gaussian fermionic PEPS. We begin with a fermionic free matter state with global symmetry, in accordance with the usual matter formulations in gauge theories – and thus we will construct a fermionic PEPS (fPEPS) (for non fermionic lattice gauge PEPS, see [33] – with matter, and [34] – without).

Within the class of fermionic PEPS, the states we would like to construct and then gauge will be Gaussian fermionic PEPS [35]. Gaussian states are ground states of quadratic, free, non-interacting Hamiltonians, and thus gauging them is analogous to gauging a free matter theory through the conventional Hamiltonian (or Lagrangian) procedure. They are fully described in terms of their second moments (covariance matrix) – expectation values of fermionic bilinear, quadratic operators; all other correlation functions are obtainable using the Wick theorem [36]. Therefore, they make sense as a starting point from both physical and computational grounds.

Once again, we will focus on the $U(1)$ case in $2+1$ dimensions, as it captures all the important qualitative features without mathematical complications that have to do with choosing some more complicated gauge group or higher dimensions. This derivation is discussed in detail in [13]. The $SU(2)$ generalization may be found in [15], while a discussion for general gauge groups and dimensions is in [18].

We will hence begin with the construction of globally invariant fPEPS, and then turn to gauging them.

## 3.1    The Physical Setting

The physical ingredients we would like to include have already been discussed: a single fermionic mode, created by $\psi^\dagger(\mathbf{x})$, is defined at each vertex. We will use a staggered fermionic formulation as before, but a slightly different one, which is obtained from the previous one by a particle-hole transformation of the odd sublattice. Previously, the absence of a fermion on an odd vertex represented the presence of an anti-particle. Here, the presence of a fermion will represent the presence of an anti-particle. This will allow us to define things in a more translationally invariant way on the level of states. The local fermionic charges are now

$$Q\left(\mathbf{x}\right) = (-1)^{\mathbf{x}}\,\psi^\dagger\left(\mathbf{x}\right)\psi\left(\mathbf{x}\right)\tag{67}$$

and the Hamiltonian $H_f$ discussed above will transform (up to a constant) to the superconducting form,

$$H_f = M \sum_{\mathbf{x}} \psi^\dagger (\mathbf{x}) \psi (\mathbf{x}) + \epsilon \sum_{\mathbf{x},i=1,2} (-1)^{\mathbf{x}} \left( \psi^\dagger (\mathbf{x}) \psi^\dagger (\mathbf{x} + \hat{\mathbf{e}}_i) + h.c. \right). \tag{68}$$

We wish to construct a state $|\psi_0\rangle$ with a global $U(1)$ invariance; that is, $|\psi_0\rangle$ should be annihilated by

$$\mathcal{Q} = \sum_{\mathbf{x}} Q(\mathbf{x}) = \sum_{\mathbf{x}} (-1)^{\mathbf{x}} \psi^\dagger (\mathbf{x}) \psi (\mathbf{x}) \tag{69}$$

or invariant under transformations generated by it,

$$e^{i\Lambda\mathcal{Q}} |\psi_0\rangle = |\psi_0\rangle. \tag{70}$$

One can also demand further symmetries, such as translation and (lattice) rotation invariance, which we will discuss later on.

## 3.2   The Local Ingredients

Naively, we would first like to construct the states $\big|A(\mathbf{x})\big\rangle$, and their product $\otimes_{\mathbf{x}} \big|A(\mathbf{x})\big\rangle$. However, a product state of fermions is not well-defined, since a tensor product factorization of a fermionic Fock space is not well defined. One could define some ordering but we would like to avoid that. Instead, at each vertex we will deal with an operator $A(\mathbf{x})$, involving both the physical and virtual degrees of freedom, and construct the product state as

$$|A\rangle = \prod_{\mathbf{x}} A(\mathbf{x}) |\Omega\rangle \tag{71}$$

where $|\Omega\rangle$ is some initial state of the system – including the Fock vacuum of the physical fermions.

A product of fermionic operators is still not well defined without specifying some lattice ordering, unless all the $A(\mathbf{x})$ operators commute. As each of them involves degrees of freedom defined on a different vertex, they will commute if and only if each $A(\mathbf{x})$ has an even fermionic parity, which we shall thus demand. In order to do that, the virtual degrees of freedom must be represented by fermions as well, and we simply interpret $|\Omega\rangle$ as the total Fock vacuum, of both physical and virtual modes (otherwise, we must have an even local *physical* parity at each vertex, which does not make any physical sense; moreover, in our current example, with only one physical mode per vertex, even parity with non-fermionic virtual modes implies that the physical fermions cannot be excited).

Now we can focus on a single vertex. In the example we construct in the following, besides the physical mode $\psi^\dagger$ we introduce eight virtual (auxiliary) fermionic modes, created by $r_\pm^\dagger, u_\pm^\dagger, l_\pm^\dagger, d_\pm^\dagger$, associated with the four legs attached to the vertex as before (right, up, left, down). Out of these, we construct the virtual electric fields,

$$E_0 (\mathbf{x}, z) = (-1)^{\mathbf{x}} \left( z_+^\dagger (\mathbf{x}) z_+ (\mathbf{x}) - z_-^\dagger (\mathbf{x}) z_- (\mathbf{x}) \right) \tag{72}$$

where $z = r, u, l, d$, and the *virtual Gauss law operator*

$$\mathcal{G}_0 \equiv E_0(r) + E_0(u) - E_0(l) + E_0(d) - \psi^\dagger \psi \tag{73}$$

– note that the staggering was incorporated into the definition of $E_0(z, \mathbf{x})$.

The most general Gaussian state $|A\rangle$ will be created from the Fock vacuum $|\Omega\rangle$ using operators of the form

$$A(\mathbf{x}) = \exp\left( \hat{T}_{ij}(\mathbf{x}) \alpha_i^\dagger(\mathbf{x}) \alpha_j^\dagger(\mathbf{x}) \right) \tag{74}$$

where $\left\{ \alpha_i^\dagger \right\}$ include all the possible creation operators, either physical or virtual.

We wish to construct $A(\mathbf{x})$ which are invariant under this virtual Gauss law operator,

$$e^{i\Lambda(\mathbf{x})\mathcal{G}_0(\mathbf{x})} A(\mathbf{x}) e^{-i\Lambda(\mathbf{x})\mathcal{G}_0(\mathbf{x})} = A(\mathbf{x}) \tag{75}$$

In order to do that, let us sort our modes to positive and negative ones, depending on the sign of the phase which is put on them by this transformation. The negative ones, $\left\{a_i^\dagger\right\}$ are the ones whose number operator appears in $\mathcal{G}_0$ with a minus sign on an even vertex, and therefore

$$e^{i\Lambda\mathcal{G}_0}a_i^\dagger e^{-i\Lambda\mathcal{G}_0} = e^{-i\Lambda}a_i^\dagger, \tag{76}$$

while the positive ones, $\left\{b_i^\dagger\right\}$ are represented by number operators with a plus sign in front, giving rise to

$$e^{i\Lambda\mathcal{G}_0}b_i^\dagger e^{-i\Lambda\mathcal{G}_0} = e^{i\Lambda}b_i^\dagger. \tag{77}$$

On an even vertex, the negative modes are $\left\{a_i^\dagger\right\} = \left\{\psi^\dagger, r_-^\dagger, u_-^\dagger, l_+^\dagger, d_+^\dagger\right\}$ and the positive ones - $\left\{b_i^\dagger\right\}= \left\{r_+^\dagger, u_+^\dagger, l_-^\dagger, d_-^\dagger\right\}$. On the odd sublattice, $\left\{a_i^\dagger\right\} = \left\{\psi^\dagger, r_+^\dagger, u_+^\dagger, l_-^\dagger, d_-^\dagger\right\}$ and the positive ones - $\left\{b_i^\dagger\right\}= \left\{r_-^\dagger, u_-^\dagger, l_+^\dagger, d_+^\dagger\right\}$.

It is easy to convince ourselves that the symmetry condition is satisfied if and only if the constructed state has the form

$$A\left(\mathbf{x}\right) = \exp\left(T_{ij}\left(\mathbf{x}\right)a_i^\dagger\left(\mathbf{x}\right)b_j^\dagger\left(\mathbf{x}\right)\right), \tag{78}$$

where $T_{ij}$ is a $5\times 4$ matrix; the first row corresponds to the coupling virtual and physical modes, and the remaining square block, $\tau$ couples the virtual modes among themselves. A minimal choice is $T_{ij}\left(\mathbf{x}\right) = T_{ij}$. Out of it we can construct a translationally invariant fPEPS (that will become charge conjugation invariant once we gauge): the staggering implies that when one translates by one lattice site, $a_i^\dagger \longleftrightarrow b_i^\dagger$ for the virtual modes (translation by one lattice site for staggered fermions is charge conjugation), and thus the square, virtual block $\tau$ has to be antisymmetric.

We obtain, for every $\mathbf{x}$, that

$$e^{i\Lambda\mathcal{G}_0}Ae^{-i\Lambda\mathcal{G}_0} = W_r W_u W_l^\dagger W_d^\dagger V^\dagger AV W_d W_l W_u^\dagger W_r^\dagger \tag{79}$$

– the PEPS condition for global symmetry, as in the non-fermionic case.

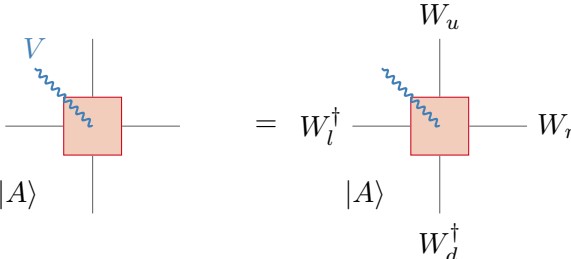

## 3.3  Contracting the Globally Invariant fPEPS

Once again, to avoid the product of fermionic states, we use operators instead of states for the contraction. Since we wish our final state to be Gaussian, we will use Gaussian projectors for that. On a horizontal link, we define the (unnormalized) projector

$$\omega_{\mathbf{x},1} = \exp\left(l_+^\dagger\left(\mathbf{x}+\hat{\mathbf{e}}_1\right)r_-^\dagger\left(\mathbf{x}\right) + l_-^\dagger\left(\mathbf{x}+\hat{\mathbf{e}}_1\right)r_+^\dagger\left(\mathbf{x}\right)\right)\Omega_\ell \exp\left(r_-\left(\mathbf{x}\right)l_+\left(\mathbf{x}+\hat{\mathbf{e}}_1\right) + r_+\left(\mathbf{x}\right)l_-\left(\mathbf{x}+\hat{\mathbf{e}}_1\right)\right) \tag{80}$$

and on a vertical one,

$$\omega_{\mathbf{x},2} = \exp\left(u_+^\dagger\left(\mathbf{x}\right)d_-^\dagger\left(\mathbf{x}+\hat{\mathbf{e}}_2\right) + u_-^\dagger\left(\mathbf{x}\right)d_+^\dagger\left(\mathbf{x}+\hat{\mathbf{e}}_2\right)\right)\Omega_\ell \exp\left(d_-\left(\mathbf{x}+\hat{\mathbf{e}}_2\right)u_+\left(\mathbf{x}\right) + d_+\left(\mathbf{x}+\hat{\mathbf{e}}_2\right)u_-\left(\mathbf{x}\right)\right), \tag{81}$$

where in both cases $\Omega_\ell$ is the projector to the vacuum states of the modes on the link's edges.

It is easy to see that for horizontal links

$$W_l W_r \omega_\ell = \omega_\ell = \omega_\ell W_r W_l \tag{82}$$

while for vertical ones

$$W_u W_d \omega_\ell = \omega_\ell = \omega_\ell W_d W_u \tag{83}$$

– this can be recognized as the usual projectors symmetry conditions for a PEPS, but different than the ones previously used, due to the staggering.

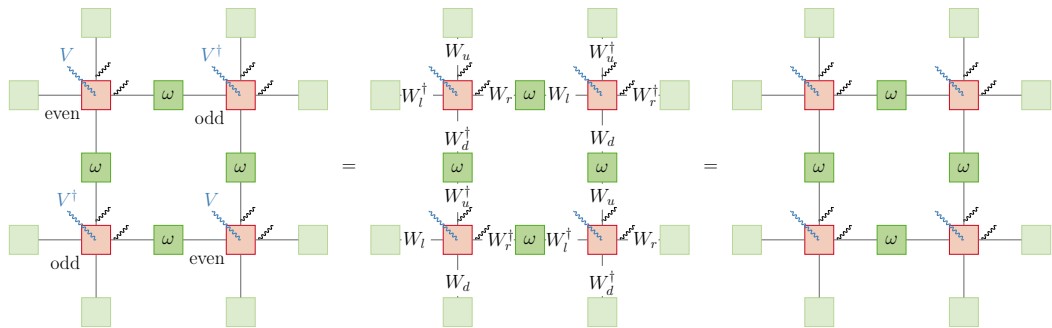

The fPEPS is then given by

$$|\psi_0\rangle = \langle \Omega_v | \underbrace{\prod_\ell \omega_\ell}_{\text{Step 4}} \underbrace{\prod_{\mathbf{x}} A(\mathbf{x})}_{\text{Step 1}} |\Omega\rangle, \tag{84}$$

where $|\Omega_v\rangle$ is the virtual vacuum. The global transformations take the form $e^{i\Lambda \mathcal{Q}} = \prod_{\mathbf{x}\,\text{even}} V(\mathbf{x}) \prod_{\mathbf{x}\,\text{odd}} V^\dagger(\mathbf{x})$ – but as this staggering has already been taken care of, $e^{i\Lambda \mathcal{Q}} |\psi_0\rangle = |\psi_0\rangle$.

To avoid confusing notations due to the staggering, we show this only graphically:

## 3.4 Gauging the PEPS

We go on to gauging the state $|\psi\rangle$. We introduce gauge field Hilbert spaces on the links, but how shall we relate them to the fermions? We simply have to modify the gauged PEPS we constructed previously to the fermionic case:

$$|\psi\rangle = \langle \Omega_v | \underbrace{\prod_\ell \omega_\ell}_{\text{Step 4}} \underbrace{\prod_\ell \mathcal{U}_G(\ell)}_{\text{Step 3}} \underbrace{\prod_{\mathbf{x}} A(\mathbf{x})}_{\text{Step 1}} |0\rangle_{\mathbf{x}} |\Omega\rangle, \tag{85}$$

where $\mathcal{U}_G(\ell)$ is a gauging transformation that entangles the virtual fermion at the beginning of the link $\ell$ with the physical gauge field we introduce on it, i.e. it rotates the virtual fermion with respect to the gauge field. It is defined as

$$\mathcal{U}_G(\ell) = \begin{cases} \int d\phi\, e^{i(-1)^{\mathbf{x}} \phi(\mathbf{x},1) E_0(r,\mathbf{x})} |\phi\rangle \langle \phi|_\ell & \ell \text{ horizontal} \\ \int d\phi\, e^{i(-1)^{\mathbf{x}} \phi(\mathbf{x},2) E_0(u,\mathbf{x})} |\phi\rangle \langle \phi|_\ell & \ell \text{ vertical.} \end{cases} \tag{86}$$

We define

$$A_G(\mathbf{x}) = \mathcal{U}_G(\mathbf{x},1)\,\mathcal{U}_G(\mathbf{x},2)\,A(\mathbf{x})\,|0\rangle_{\mathbf{x}}. \tag{87}$$

What are the symmetry conditions satisfied by this object? It is easy to see that for an even vertex $\mathbf{x}$,

$$V_{r,u} A_G(\mathbf{x}) |\Omega\rangle = W_{r,u} A_G(\mathbf{x}) |\Omega\rangle \tag{88}$$

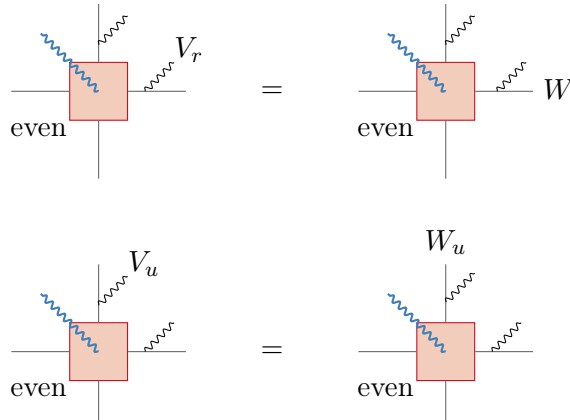

while for an odd one

$$V_{r,u} A_G(\mathbf{x}) |\Omega\rangle = W_{r,u}^\dagger A_G(\mathbf{x}) |\Omega\rangle \tag{89}$$

– the properties that allow us to gauge, just like in (30). From these two equations, and the symmetry condition of $A(\mathbf{x})$, we obtain that for an even vertex,

$$V_r V_u V^\dagger A_G(\mathbf{x}) |\Omega\rangle = W_l W_d A_G(\mathbf{x}) |\Omega\rangle \tag{90}$$

while for an odd one,

$$V_r V_u V A_G(\mathbf{x}) |\Omega\rangle = W_l^\dagger W_d^\dagger A_G(\mathbf{x}) |\Omega\rangle \tag{91}$$

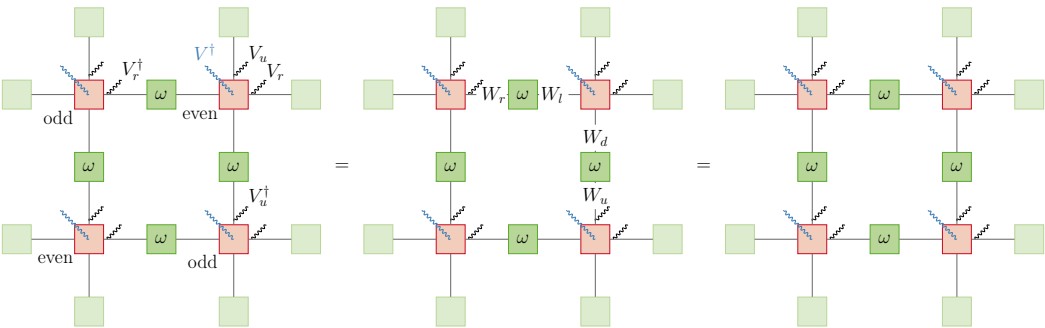

This allows us to verify that $|\psi\rangle$ is gauge invariant, i.e. that for an arbitrary $\mathbf{x}_0$,

$$e^{i\Lambda\mathcal{G}(\mathbf{x}_0)} |\psi\rangle = V_r(\mathbf{x}_0) V_u(\mathbf{x}_0) V_l^\dagger(\mathbf{x}_0 - \hat{\mathbf{e}}_1) V_d^\dagger(\mathbf{x}_0 - \hat{\mathbf{e}}_2) V^\dagger(\mathbf{x}_0) |\psi\rangle$$
$$= |\psi\rangle \tag{92}$$

Once again we show it pictorially, for the two possible parities of the vertex around which the state is transformed. First, even,

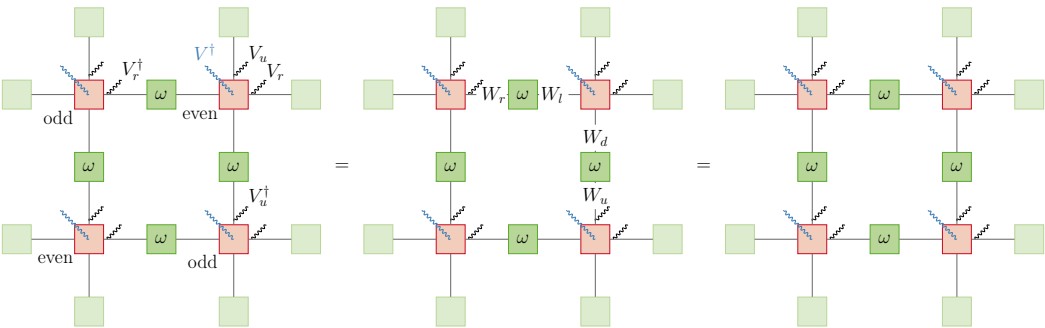

and finally odd,

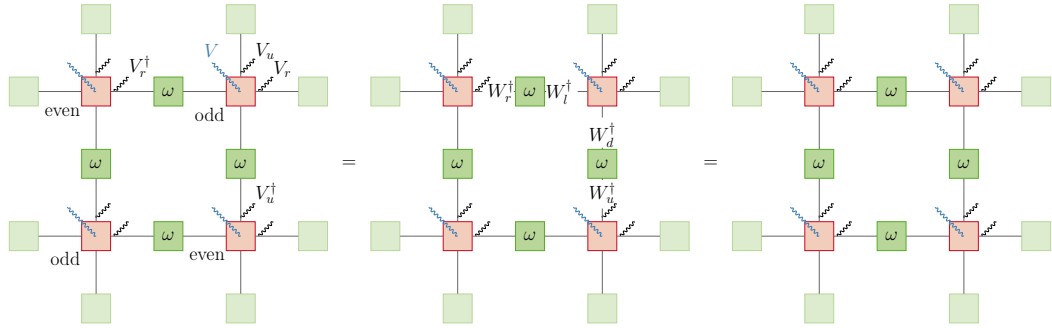

Another thing that has to be mentioned, is that in this method the physical electric field is truncated, and cannot take any integer eigenvalue, but rather only, in this case, $0, \pm 1$. This is once again due to the fact that in the gauging process (the action of $\mathcal{U}_G$) transfers the state of the virtual electric field to the physical one, and the virtual electric fields, constructed out of fermions in this particular manner (as the difference between two fermionic number operators) only take these values. In order to increase the truncation, one has to add more virtual fermions on each leg, such that higher eigenvalues of $E_0$ are obtained. This is called, in usual PEPS terms, increasing the *bond dimension*, and it also allows one, in general, to increase the number of parameters the PEPS depends on.

## 3.5 Other Symmetries

So far, we have constructed a fermionic Gaussian PEPS with a global $U(1)$ symmetry, and made it local using our gauging procedure. What can we say about other symmetries obeyed by the state?

We have already commented how to encode translational invariance into the globally invariant state $|\psi_0\rangle$, and that a translation by one lattice site, which is, for staggered fermions, charge conjugation, requires exchanging $a_i^\dagger \longleftrightarrow b_i^\dagger$. If we consider the staggering in the definition of $\mathcal{U}_G(\ell)$, we see that the transformation of the gauge field that leaves $|\psi\rangle$ invariant is translation + inversion of sign,

$$\phi(\mathbf{x}, i) \longrightarrow -\phi(\mathbf{x} + \hat{\mathbf{e}}_j, i) \tag{93}$$

– which is exactly charge conjugation! Of course, it makes no sense to talk about simple translational invariance when we discuss staggered fermions, where the unit cell includes more than one vertex. If we consider translation by two sites, on the other hand, we have simple translational invariance,

$$\phi(\mathbf{x}, i) \longrightarrow \phi(\mathbf{x} + 2\hat{\mathbf{e}}_j, i) \tag{94}$$

as expected for such a staggered system.

Another symmetry that could be considered is rotation. First of all, for the globally invariant state without the gauge field. For that, one has to define the right transformation properties of all fermions under the rotation. This involves not only changing the coordinate respectively, but also requires more. As we discuss fermions, a complete rotation of the physical fermion must put a sign on it. Therefore, the basic $\pi/2$ rotation, realized by an operator $\mathcal{U}_R$, must give rise to

$$\mathcal{U}_R \psi^\dagger(\mathbf{x}) \mathcal{U}_R^\dagger = \eta_p \psi^\dagger(R\mathbf{x}) \tag{95}$$

with $\eta_p^4 = -1$; $R\mathbf{x}$ is the rotated coordinate.

The virtual modes around each vertex must be exchanged by such a transformation, up to possible phases, keeping positive modes positive and negative modes negative. That is, we need to have, for the virtual components,

$$\begin{aligned}
\mathcal{U}_R a_i^\dagger(\mathbf{x}) \mathcal{U}_R^\dagger &= M_{ij}^A a_j^\dagger(R\mathbf{x}) \\
\mathcal{U}_R b_i^\dagger(\mathbf{x}) \mathcal{U}_R^\dagger &= M_{ij}^B b_j^\dagger(R\mathbf{x})
\end{aligned} \tag{96}$$

with rotation matrices that contain only one nonzero element per row and column, and that has to be a phase. The phase must be chosen such that the product of projectors $\prod_\ell \omega_\ell$ is invariant.

After gauging, one simply has to add the rotation rules for the gauge field. Once these two requirements (translation and rotation invariance) are taken care of, and we remove redundancies having to do with phase symmetries of the virtual fermions in which the physical fermions do not participate. We arrive at a final characterization [13],

$$T = \begin{pmatrix}
t & \eta_p^2 t & \eta_p t & \eta_p^3 t \\
0 & y & z/\sqrt{2} & z/\sqrt{2} \\
-y & 0 & -z/\sqrt{2} & z/\sqrt{2} \\
-z/\sqrt{2} & z/\sqrt{2} & 0 & y \\
-z/\sqrt{2} & -z/\sqrt{2} & -y & 0
\end{pmatrix}, \tag{97}$$

where $t \geq 0$ and $y, z \in \mathbb{C}$. $t = 0$ is of course meaningless in the global case, but in the local case it gives rise to pure gauge states.

## 3.6 Generalizations

One can generalize the gauging prescription for PEPS, and in particular Gaussian fPEPS, described above, in the following ways:

1. *Different spatial dimensions*: one has to introduce more similar virtual legs, and apply the gauging transformation only in the outgoing directions. A similar procedure holds for other geometries too.

2. *Different gauge groups*: similar procedure applies to further gauge groups (see [15] for an explicit $SU(2)$ construction, and [18] for a general formulation). In general, for non-Abelian groups, the symmetry conditions become more complicated technically, as there is difference between left and right group transformations [12,23] (the electric field on the link differs from one edge to another – it has both *left* and *right* fields, whose difference manifests the fact that a non-Abelian gauge fields, unlike Abelian ones, carry charge – e.g. gluons vs. photons).

3. *Larger bond dimensions*: as explained before, one may add more virtual fermions in order to enlarge the dimension of the virtual Hilbert spaces and introduce more parameters on which the state depends. This, however, must be done carefully in a way that respects the symmetry (only complete irreducible multiplets can be included – and also multiple copies thereof). This increases the truncation of the virtual Hilbert spaces, and hence of the physical Hilbert space on the links as well (electric field truncation).

## 3.7 Analytical (Toy Model) Results

Which type of physics is manifested by "toy" states, with the minimal bond dimension (truncated electric fields), as described above?

One way to study that analytically is to put the system on a cylinder – with one compact and one open dimension – and define an object called the PEPS transfer matrix, similar to transfer matrix in statistical mechanics [37]. The transfer matrix "transfers" information from layer to layer of the cylindrical system, along the open direction. If the spectrum of the transfer matrix is gapped, correlation functions with respect to the PEPS show an exponential decay.

A conventional way to study such PEPS is therefore to compute the lowest eigenvalues of the transfer matrix for some different choices of the PEPS parameters, and see whether it is gapped or not – implying that the PEPS could be the ground state of a gapped Hamiltonian, or not. This allows one to draw a "phase diagram of states" in this parameter space – that is, a phase diagram for the set of Hamiltonians of which the set of PEPS obtained by varying the parameters, are ground states (usually referred to as parent Hamiltonians). Parameter space surfaces on which the transfer matrix is gapless are suspected as phase boundaries.

One could then draw a phase diagram in this method, and calculate the expectation values of some significant physical operators – order parameters – within the phases (gapped regions). Such studies, with Wilson loops [1], for $U(1)$ states [13], have shown both confining and non-confining gapped phases for static charges, in the pure gauge case. Similar $SU(2)$ studies [15] have shown a gapped deconfining phase as well as a gapless phase with perimeter-law decay of the area law – a hint of a possible Higgs phase [38].

Another way to detect possible phase transitions is to look for virtual symmetries: symmetries in which the physical degrees of freedom do not participate. In the $SU(2)$ case [15], for example, such a particle-hole symmetry exists for the virtual fermions. This symmetry allows one to construct a parameter space transformation that connects states which are physically identical but constructed using different parameters. The fixed lines of this transformation are reasonably suspected as phase boundaries – and indeed, when those were compared to transfer matrix calculations, they gave rise to the same results.

## 3.8 Contracting Gauged Gaussian Fermionic PEPS: Combining Monte-Carlo with Tensor Network Methods

In many cases, tensor network states are used as ansatz states in variational calculations. While this has been done quite successfully for one dimensional systems, with DMRG (density matrix renormalization group) techniques [39], the higher dimensional generalizations are problematic as they do not scale well [10].

From the lattice gauge theory point of view, the widely used Monte-Carlo methods suffer from the sign problem [5] in several physically interesting scenarios (e.g. finite chemical potential). They also do not allow one to consider directly unitary time evolution, as they are formulated in a Euclidean spacetime.

Indeed, in one space dimension, tensor network numerical studies have shown remarkable results, even in otherwise problematic scenarios [27, 40–62]. However, what about higher dimensional systems?

Gaussian fermionic PEPS suggest a way to circumvent these problems. As we shall see, one may calculate, using Monte-Carlo methods, the expectation values of physical observables with respect to gauged Gaussian PEPS, exploiting the simple and efficient Gaussian formalism, based on Gaussian integration and matrix operations [36]. Such calculations do not depend on the dimension of the system, overcoming the first problem. They also do not encounter the sign problem, as they are carried out in the Hamiltonian, Hilbert space formulation (expectation values of physical operators) and not in the Wick rotated path integral formulation that encounters it.

For a general discussion see [18]; here, we shall briefly review the simple $U(1)$ case.

We have already mentioned that the Hilbert space of a fermionic lattice gauge theory $\mathcal{H}$, with no static charges,

could be embedded in the product space

$$\mathcal{H} \subset \mathcal{H}_{\mathrm{g}} \times \mathcal{H}_{\mathrm{f}}, \tag{98}$$

where $\mathcal{H}_{\mathrm{g}}$ is the gauge field Hilbert space, covering all links, and $\mathcal{H}_{\mathrm{f}}$ is the fermionic Fock space, covering all vertices. The most general state in this Hilbert space may be expanded as

$$|\psi\rangle = \int \mathcal{D}\Phi \, |\Phi\rangle \, |\psi(\Phi)\rangle, \tag{99}$$

where $|\Phi\rangle = \underset{\ell}{\otimes} |\phi_\ell\rangle \in \mathcal{H}_{\mathrm{g}}$ is some gauge field configuration state on all the links, and $\mathcal{D}\Phi = \prod_\ell d\phi_\ell$. $|\psi(\Phi)\rangle \in \mathcal{H}_{\mathrm{f}}$ is a general fermionic state, that depends on the gauge field configuration as a parameter, and is in general not normalized. The state $|\psi\rangle$ could be very general, not necessarily a PEPS and even not manifesting any symmetry; of course, we are interested in the case when it is gauge invariant. Then, $|\psi(\Phi)\rangle$ is a state of fermions coupled to an external, static $U(1)$ field with configuration $\Phi$.

When $|\psi\rangle$ is a gauged Gaussian fermionic PEPS, the states $|\psi(\Phi)\rangle$ are merely fermionic Gaussian states, and therefore admit a very simple description and analytical treatment using the Gaussian formalism [36]. This allows us to efficiently calculate the expectation values of physical operators for them.

Consider the (square) norm of the state. As the configuration states $|\Phi\rangle$ form a complete set, it can be simply expressed as

$$\langle\psi|\psi\rangle = \int \mathcal{D}\Phi \, \langle\psi(\Phi)|\psi(\Phi)\rangle \equiv Z \tag{100}$$

– an integral over squares of norms of Gaussian states, $\langle\psi(\Phi)|\psi(\Phi)\rangle$, whose computation is simple and numerically efficient. They are all positive definite, and thus the function

$$p(\Phi) = \langle\psi(\Phi)|\psi(\Phi)\rangle /Z \tag{101}$$

is a probability density function in the space of gauge field configurations, and $Z$ is the partition function associated with it.

Now we are ready to compute expectation values. Consider first the Wilson loop [1], an oriented product of group element operators along some closed path $\mathcal{C}$:

$$W(\mathcal{C}) = \prod_{\ell \in \mathcal{C}} U(\ell), \tag{102}$$

where $\ell$ is now an oriented link. We define orientation with respect to the Gauss law. If the link is following an outgoing leg (up, right), it is considered *forward*. The other two links (down, left) are *backward*. If its orientation is backwards one has to use $U^\dagger$ on that link, such that the operator $W(\mathcal{C})$ is gauge invariant. The configuration states are eigenstates of $W(\mathcal{C})$,

$$W(\mathcal{C}) |\Phi\rangle = \exp\left(i \sum_{\ell \in \mathcal{C}} \phi(\ell)\right) |\Phi\rangle \tag{103}$$

where now a minus sign is understood in front of the phase of a backward link. We therefore obtain that

$$\langle W(\mathcal{C})\rangle = \langle\psi| W(\mathcal{C}) |\psi\rangle /Z = \int \mathcal{D}\Phi p(\Phi) \exp\left(i \sum_{\ell \in \mathcal{C}} \phi(\ell)\right), \tag{104}$$

– which can be computed using Monte-Carlo.

Let us consider another type of physical operator – a mesonic string, an open gauge field string enclosed by fermionic operators, e.g.

$$M(\mathbf{x}, \mathbf{y}, \mathcal{C}) = \psi^\dagger(\mathbf{x}) \prod_{\ell \in \mathcal{C}} U(\ell) \, \psi(\mathbf{y}) \tag{105}$$

where now $\mathcal{C}$ is an open path connecting the endpoints $\mathbf{x}, \mathbf{y}$. In this case, we obtain that

$$\langle M(\mathbf{x}, \mathbf{y}, \mathcal{C})\rangle = \int \mathcal{D}\Phi p(\Phi) \, e^{i \sum_{\ell \in \mathcal{C}} \phi(\ell)} \frac{\langle\psi(\Phi)| \psi^\dagger(\mathbf{x}) \psi(\mathbf{y}) |\psi(\Phi)\rangle}{\langle\psi(\Phi)|\psi(\Phi)\rangle} \tag{106}$$

– this integral involves the expectation value of a quadratic fermionic operator with respect to a fermionic Gaussian state – $\left\langle \psi^\dagger\left(\mathbf{x}\right)\psi\left(\mathbf{y}\right)\right\rangle_\Phi = \frac{\left\langle \psi(\Phi)\left|\psi^\dagger(\mathbf{x})\psi(\mathbf{y})\right|\psi(\Phi)\right\rangle}{\left\langle \psi(\Phi)|\psi(\Phi)\right\rangle}$ – which is very simple to calculate in the Gaussian formalism using covariance matrix elements of $\left|\psi\left(\Phi\right)\right\rangle$. Therefore, $\left\langle M\left(\mathbf{x},\mathbf{y},\mathcal{C}\right)\right\rangle$ can be calculated too using Monte Carlo integration of quantities obtained from Gaussian calculations.

Similar results are obtained for other operators – such as ones involving electric fields [18]. In all cases, the quantum nature of the gauge field Hilbert space is handled before the integration takes place, leaving traces of the gauge field only in the form of integration variables. The quantum nature of the fermionic part is taken care of through the Gaussian formalism, thanks to which everything can be computed efficiently – as functions of the gauge field configurations. Eventually one is left with a $\int\mathcal{D}\Phi$ integral over the gauge field configurations that can be computed with Monte-Carlo. This opens the way for using such states, with higher bond dimensions, as varitational guesses for ground states of lattice gauge theory Hamiltonians, for example.

# 4 The Opposite Process: Eliminating the Fermions

## 4.1 Solving Gauss Law for the Matter Field

We have argued that the Gauss law (43) is problematic to solve for the gauge field, since it is a differential equation with no unique solution, if the spatial dimension is more than one (and in the one dimensional case, the solution is unique but nonlocal). However, note that if we see it as an equation for the matter, the setting is completely different: then, it is a very simple algebraic equation, in which the charge (or its density) – not a vector quantity – is already explicitly and uniquely given as a function of the divergence of the electric field.

Therefore, in cases where each local charge $Q\left(\mathbf{x}\right)$ eigenstate is obtained uniquely by a single configuration of the matter degree of freedom at $\mathbf{x}$, one could indeed write down an "opposite" unitary transformation; that will take a gauge invariant state or Hamiltonian, with both matter and gauge fields as quantum degrees of freedom, and decouple the matter using the Gauss laws. This produces a state, or a Hamiltonian, without the symmetry, but also without the Gauss law constraints; with the same physical information, and effectively less degrees of freedom. It is, in fact, a controlled operation that changes the state of the matter at each vertex, controlled by the values of the electric field on the links around it. It is a well known procedure for Higgs matter fields, known as fixing the *unitary gauge*.

## 4.2 The Unitary Gauge of Higgs Theories

Let us consider a lattice formulation of the Higgs mechanism, similar to that of [38]. On each vertex, we have a complex scalar field $\Phi\left(\mathbf{x}\right)$ which may be expanded in terms of two bosonic modes. We will write it in a polar form,

$$\Phi\left(\mathbf{x}\right) = R\left(\mathbf{x}\right)e^{i\theta\left(\mathbf{x}\right)}. \tag{107}$$

In the usual quasi-classical treatment of the Higgs mechanism, $R\left(\mathbf{x}\right)$ is the Higgs field, and $\theta\left(\mathbf{x}\right)$ is the Goldstone mode, which are independent degrees of freedom, satisfying

$$\left[R\left(\mathbf{x}\right),\theta\left(\mathbf{y}\right)\right] = 0. \tag{108}$$

The phase $\theta\left(\mathbf{x}\right)$ is canonically conjugate to the local charge $Q\left(\mathbf{x}\right)$, having a non bounded integer spectrum, raised by $e^{i\theta\left(\mathbf{x}\right)}$, and completely decoupled from the radial degree of freedom $R\left(\mathbf{x}\right)$. In [38], the Higgs field is frozen – $R\left(\mathbf{x}\right) = R_0$, and we will do it too, as it has no relevance for us. On the links, we have the usual $U\left(1\right)$ gauge field Hilbert space introduced above (note that in this case, the link and matter Hilbert spaces are the same).

The interaction Hamiltonian between the matter and gauge fields takes the form

$$\widetilde{H} = \epsilon\sum_{\mathbf{x},i=1,2}\left(e^{i\theta\left(\mathbf{x}\right)}U\left(\mathbf{x},i\right)e^{-i\theta\left(\mathbf{x}+\hat{\mathbf{e}}_i\right)} + h.c.\right) = 2\epsilon\sum_{\mathbf{x},i=1,2}\cos\left(\theta\left(\mathbf{x}\right) + \phi\left(\mathbf{x},i\right) - \theta\left(\mathbf{x}+\hat{\mathbf{e}}_i\right)\right). \tag{109}$$

As usual, in the Higgs mechanism discussion, we fix the gauge to the *unitary gauge* [38]: perform a pure-gauge transformation (one that only acts on the gauge field degrees of freedom), in which the Goldstone modes are absorbed by the gauge field, making it massive. The transformation is

$$\phi\left(\mathbf{x},i\right) \longrightarrow \phi\left(\mathbf{x},i\right) - \theta\left(\mathbf{x}\right) + \theta\left(\mathbf{x}+\hat{\mathbf{e}}_i\right). \tag{110}$$

How is it implemented quantum mechanically? It is a straightforward exercise to convince ourselves that it takes the form

$$\mathcal{U}_H = \prod_{\mathbf{x}} \mathcal{U}_H(\mathbf{x})$$
$$\mathcal{U}_H(\mathbf{x}) = e^{iG(\mathbf{x})\theta(\mathbf{x})} \tag{111}$$
$$= e^{i\left(E(\mathbf{x},1)+E(\mathbf{x},2)-E(\mathbf{x}-\hat{\mathbf{e}}_1,1)-E(\mathbf{x}-\hat{\mathbf{e}}_2,2)\right)\theta(\mathbf{x})}$$

– where, by expressing it as a vertex-dependent transformation, we changed the point of view: instead of transforming the gauge field with respect to the matter configuration, we look at it now as a transformation that changes the matter field on each vertex, controlled by the electric fields on the links around it. It changes the matter charge at each vertex exactly by the amount of electric field divergence, and thanks to the Gauss law it means that it is reduced to zero. This is the controlled operation we wanted for decoupling the matter (while eliminating the gauge constraints)! If we act with it on a gauge invariant state in this Hilbert space, we will get a product of a trivial matter state with a gauge field superposition that still contains the same physical information, but without the symmetry: for an arbitrary gauge invariant state,

$$|\psi\rangle = \sum_{\{Q(\mathbf{x})\}} \alpha\left(\{Q(\mathbf{x})\}\right) \left|E\left(\{Q(\mathbf{x})\}\right)\right\rangle_{\text{Gauge}} \left|\{Q(\mathbf{x})\}\right\rangle_{\text{Matter}} \tag{112}$$

with a Gauss law at each vertex, we have that

$$\mathcal{U}_H |\psi\rangle = \left|\{Q(\mathbf{x})=0\}\right\rangle_{\text{Matter}} \otimes \sum_{\{Q(\mathbf{x})\}} \alpha\left(\{Q(\mathbf{x})\}\right) \left|E\left(\{Q(\mathbf{x})\}\right)\right\rangle_{\text{Gauge}} \tag{113}$$

without any Gauss law constraints.

If we consider the case of a $\mathbb{Z}_2$ gauge theory with such Higgs matter, where both the gauge field and the matter are described by spin half Hilbert spaces, $\mathcal{U}_H$, is nothing but a controlled not (CNOT) operation, as discussed in [33].

## 4.3   Eliminating Fermionic Matter

Next we wish to ask – is there an analogy for fermionic matter? The answer is: yes, but carefully: fermions come with their special statistics and its complications, and thus one has to be extra cautious, and act slightly differently than what we did above. Following [19], we will review (and demonstrate for $U(1)$) how to replace the fermionic degrees of freedom by spins (hard-core bosons).

We go back now to the $U(1)$ lattice gauge theory with fermionic matter discussed in the very beginning, with interaction terms of the form

$$\widetilde{H} = \epsilon \sum_{\mathbf{x},i=1,2} \left(\psi^\dagger(\mathbf{x}) U(\mathbf{x},i) \psi(\mathbf{x}+\hat{\mathbf{e}}_i) + h.c.\right). \tag{114}$$

In an (incomplete, as we shall see) analogy to the Higgs case, we express the fermionic mode operators as

$$\psi^\dagger(\mathbf{x}) = \eta^\dagger(\mathbf{x}) c(\mathbf{x}), \tag{115}$$

where $c(\mathbf{x})$ is a fermionic Majorana mode, satisfying the Clifford algebra

$$\{c(\mathbf{x}), c(\mathbf{y})\} = 2\delta(\mathbf{x},\mathbf{y}) \tag{116}$$

and the modes $\eta^\dagger(\mathbf{x})$ have an on-site fermionic anticommutation relation,

$$\{\eta^\dagger(\mathbf{x}), \eta(\mathbf{x})\} = 1 \quad ; \{\eta^\dagger(\mathbf{x}), c(\mathbf{x})\} = 0 \tag{117}$$

and off-site bosonic commutation relations,

$$\left[\eta^\dagger(\mathbf{x}), \eta(\mathbf{y})\right] = [\eta(\mathbf{x}), \eta(\mathbf{y})] = [c(\mathbf{x}), \eta(\mathbf{y})] = 0 \quad \text{for } \mathbf{x} \neq \mathbf{y}. \tag{118}$$

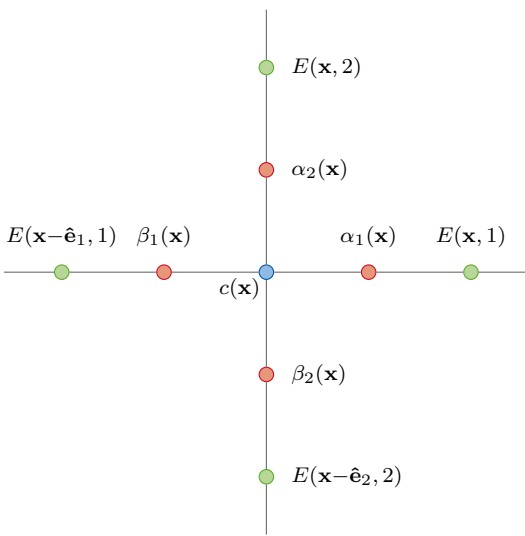

Figure 1: Visual representation of the construction detailed in (115) to (118). For clarity, we show only the auxiliary modes that are newly introduced and the electric fields. On each vertex site $\mathbf{x}$ an auxiliary fermionic Majorana mode $c(\mathbf{x})$ (blue circle) is defined. The auxiliary modes $\alpha_i(\mathbf{x})$ and $\beta_i(\mathbf{x})$ (red circles) are introduced below and used in equation (124). The electric fields on the respective links are shown as green circles.

Figure 1 shows an overview of all modes present on a vertex and its links.

The operator $\eta^\dagger(\mathbf{x})$ – almost analogous to the radial field $R(\mathbf{x})$ – behaves like the creation operator of a hard-core boson; if we can remove the operators $c(\mathbf{x})$ from the Hamiltonian, we can eventually represent $\eta^\dagger(\mathbf{x})$ by spin raising operators, $\sigma_+(\mathbf{x})$. In this sense, the operators $c(\mathbf{x})$ are analogous to $e^{i\theta(\mathbf{x})}$ in the bosonic, Higgs case; indeed, one can think about the Majorana mode operator obeying $c^2(\mathbf{x}) = 1$ as a "$\mathbb{Z}_2$ phase operator" having to do with the fermionic $\mathbb{Z}_2$ symmetry. The analogy is not complete, though; in the Higgs case, the radial and angular components were independent, commuting on-site as well, but here, although commuting for different sites, on-site we have $\{\eta^\dagger(\mathbf{x}), c(\mathbf{x})\} = 0$: the "radial" and "angular" modes are related through a $\mathbb{Z}_2$ equation. In fact, unlike in the Higgs case, the field $\eta^\dagger(\mathbf{x})$ contains all the physical information, in the sense that its charge participates in the $U(1)$ Gauss laws, while $c(\mathbf{x})$ is only related to a $\mathbb{Z}_2$ symmetry – the fermionic statistics. These two fields are related through the local $\mathbb{Z}_2$ symmetry manifested by $\{\eta^\dagger(\mathbf{x}), c(\mathbf{x})\} = 0$. Therefore, in the fermionic case the "separation", which is not absolute, between "radius" and "angle" could be seen as separating physics and statistics – held together by the anticommutation relation.

In these terms, the interaction Hamiltonian has the form

$$\widetilde{H} = \epsilon \sum_{\mathbf{x}, i=1,2} \left( \eta^\dagger(\mathbf{x}) c(\mathbf{x}) U(\mathbf{x}, i) c(\mathbf{x} + \hat{\mathbf{e}}_i) \eta(\mathbf{x} + \hat{\mathbf{e}}_i) + h.c. \right) \tag{119}$$

Ideally, we would like to have a transformation, for which

$$U(\mathbf{x}, i) \longrightarrow c(\mathbf{x}) U(\mathbf{x}, i) c(\mathbf{x} + \hat{\mathbf{e}}_i) \tag{120}$$

and we are done. This, however, would be a problem: each link is transformed from the left by $c(\mathbf{x})$ and from the right by $c(\mathbf{x} + \hat{\mathbf{e}}_i)$. This is realized with local transformations of the form

$$\mathcal{U}'_F(\mathbf{x}) = c^{G(\mathbf{x})} = c^{\left(E(\mathbf{x},1) + E(\mathbf{x},2) - E(\mathbf{x}-\hat{\mathbf{e}}_1,1) - E(\mathbf{x}-\hat{\mathbf{e}}_2,2)\right)}. \tag{121}$$

It works, thanks to the $\mathbb{Z}_2$ relation

$$e^{i\pi E} U e^{-i\pi E} = -U \tag{122}$$

responsible for

$$c^E U c^E = -cU. \tag{123}$$

However, these transformations, that "rotate $U(\mathbf{x}, i)$ by a fermionic phase", do not have a fixed fermionic parity and violate the fermionic superselection locally. It can be shown that the product of all these, which is what we need, preserves fermionic parity globally, but as it is locally broken, one has to define some order for the product, that will eventually give rise to nonlocal results.

What we do, instead, is to introduce on each $\ell = (\mathbf{x}, i)$ link two Majorana modes, $\alpha_i(\mathbf{x})$ associated with its beginning and $\beta_i(\mathbf{x} + \hat{\mathbf{e}}_i)$ with its end. These operators do not appear in the original Hamiltonian at all and hence we can fix the initial state of the respective auxiliary degrees of freedom as we wish. If the original Hilbert space, without these modes, is $\mathcal{H}$, we simply embed it within $\mathcal{H} \times \Omega$, where $\Omega$ is the one dimensional Hilbert space consisting of the Fock vacuum annihilated by the link annihilation operators

$$f_i(\mathbf{x}) = \frac{1}{2}\left(\alpha_i(\mathbf{x}) - i\beta_i(\mathbf{x} + \hat{\mathbf{e}}_i)\right) \tag{124}$$

as well as the vertex ones, $\chi(\mathbf{x})$, for which

$$c(\mathbf{x}) = \chi(\mathbf{x}) + \chi^\dagger(\mathbf{x}). \tag{125}$$

The visual representation of equation (124) is shown in Figure 2.

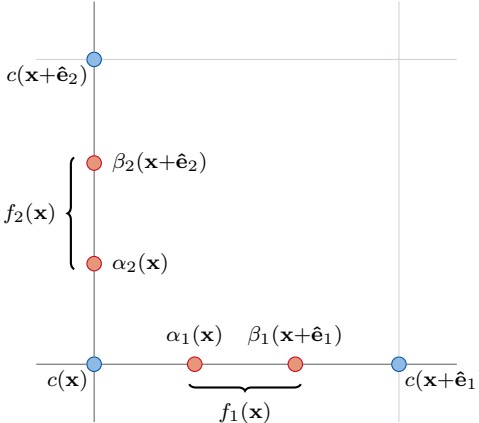

Figure 2: Pictorial representation of (124). The modes $f_i(\mathbf{x})$ related to vertex $\mathbf{x}$ in direction $i$ are formed by combining the modes $\alpha$ and $\beta$ on the respective link. The auxiliary modes $c(\mathbf{x})$ are depicted in blue as in Figure 1.

We define the local transformation

$$\mathcal{U}_F(\mathbf{x}) = \left(ic(\mathbf{x})\beta_2(\mathbf{x})\right)^{E(\mathbf{x}-\hat{\mathbf{e}}_2, 2)}\left(ic(\mathbf{x})\beta_1(\mathbf{x})\right)^{E(\mathbf{x}-\hat{\mathbf{e}}_1, 1)}\left(ic(\mathbf{x})\alpha_2(\mathbf{x})\right)^{E(\mathbf{x}, 2)}\left(ic(\mathbf{x})\alpha_1(\mathbf{x})\right)^{E(\mathbf{x}, 1)} \tag{126}$$

It has an even fermionic parity, and thus preserves the fermionic superselection rule locally; thanks to that, and having different fermionic modes at each vertex, $\left[\mathcal{U}_F(\mathbf{x}), \mathcal{U}_F(\mathbf{x}')\right] = 0$, the transformation

$$\mathcal{U}_F = \prod_{\mathbf{x}} \mathcal{U}_F(\mathbf{x}) \tag{127}$$

is well defined, since no ordering has to be specified for the product. We obtain that

$$\mathcal{U}_F \tilde{H} \mathcal{U}_F^\dagger = \epsilon \sum_{\mathbf{x}, i=1,2}\left(\xi_i \eta^\dagger(\mathbf{x})\alpha_i(\mathbf{x})U(\mathbf{x}, i)\beta_i(\mathbf{x} + \hat{\mathbf{e}}_i)\eta(\mathbf{x} + \hat{\mathbf{e}}_i) + h.c.\right), \tag{128}$$

where $\xi_i$ are phases (signs) that depend on the electric fields on some links around the link's edges [19]. Since the initial state for the auxiliary link fermions was the vacuum, and one may check that $\left[\mathcal{U}_F, \alpha_i(\mathbf{x})\beta_i(\mathbf{x} + \hat{\mathbf{e}}_i)\right] = 0$, they remain so after the transformation, and could be replaced by $-i$. The hard-core bosonic operators $\eta^\dagger(\mathbf{x})$ can

be effectively replaced by spin raising operators $\sigma_+\left(\mathbf{x}\right)$ (in a process that involves local, string-less Jordan Wigner transformations – see [19]), and we obtain a transformed interaction Hamiltonian of the form

$$-i\epsilon \sum_{\mathbf{x},i=1,2} \left(\xi_i \sigma_+\left(\mathbf{x}\right) U\left(\mathbf{x},i\right)\sigma_-\left(\mathbf{x}+\hat{\mathbf{e}}_i\right) + h.c.\right). \tag{129}$$

Other terms in the Hamiltonian also transform in a way that leaves no fermions present [19]. Therefore we see that our lattice gauge theory with fermionic matter was mapped to one with hard core bosonic matter (that can be now removed straightforwardly using the Gauss laws). The lattice rotations are broken by the $\xi_i$ operators, whose role is to keep the right statistics and commutation relations [19].

In order to demonstrate it simply, let us look at the one dimensional case, where

$$\mathcal{U}_F\left(x\right) = \left(ic\left(x\right)\beta\left(x\right)\right)^{E(x-1)}\left(ic\left(x\right)\alpha\left(x\right)\right)^{E(x)} \tag{130}$$

and one simply obtains

$$-i\epsilon \sum_x \left(e^{i\pi E(x-1)}\sigma_+\left(x\right) U\left(x\right)\sigma_-\left(x\right) + h.c.\right). \tag{131}$$

As the "radial" and "angular" degrees of freedom were related to each other, the symmetry is not broken, and a matter field still exists; however, it is not fermionic. Unlike in the Higgs case, we did not break the continuous gauge symmetry completely and did not eliminate the matter: we only exploited the $\mathbb{Z}_2$ subgroup of the gauge group, $U(1)$ in our example, to eliminate the fermionic nature of the matter. In some cases, as the one we discuss, one may move on and eliminate the hard-core bosons using a simple, non-fermionic controlled operation as the one shown above for Higgs field; the difference will be, that now the bosons are hard-core, so projectors for the right values of electric field divergence must be introduced.

In fact, this can be done for every lattice gauge theory whose gauge group includes a $\mathbb{Z}_2$ normal subgroup [19] – for example, $\mathbb{Z}_{2N}, U\left(N\right), SU\left(2N\right)$: the important property is to have an operator $E$, for which the $\mathbb{Z}_2$ relation

$$e^{i\pi E}Ue^{-i\pi E} = -U \tag{132}$$

holds. This is the basis for our transformation. Groups with a $\mathbb{Z}_2$ normal subgroup satisfy that. For other cases, e.g. $SU\left(2N+1\right)$, one can introduce an auxiliary $\mathbb{Z}_2$ gauge field – that is, extend the gauge symmetry to e.g. $SU\left(2N+1\right)\times\mathbb{Z}_2$, and use it for the elimination of fermions (and their replacement by hard core bosons) in a process as the one discussed above. It can be done in a way that preserves the physical properties of the original model (no dynamics is added for the auxiliary $\mathbb{Z}_2$ gauge field). However, if one has more than one spinor component on each site, extra caution has to be taken when representing the hard core bosons by spins: *local* Jordan-Wigner transformations must be used [19].

The relation to $\mathbb{Z}_2$ is not accidental: any "reasonable" fermionic theory without supersymmetry has a global $\mathbb{Z}_2$ symmetry – parity superselection, which we have already discussed. When we have a $\mathbb{Z}_2$ local symmetry, we can construct a transformation as above and eliminate the fermions locally since their information is "saved" by the gauge field. If there is no such field we can add it with some minimal coupling procedure and then do as above: so, in order to remove the fermionic statistics, one simply has to gauge the global $\mathbb{Z}_2$ symmetry associated with it and make it local.

When the gauge group also has a $U(1)$ normal subgroup, as in the case of $U(N)$, one can go one step further and completely eliminate the matter as in the Higgs unitary gauge [38], even in the cases of hard-core bosonic or fermionic matter [20]. As a result of this procedure, only gauge field degrees of freedom will appear, but the symmetry will be broken as in the original case; however, since the matter degrees of freedom were bounded tobegin with, some local projectors will be added, as well as local constraints - which do not appear in the Abelian Higgs case [38]. Since it is done unitarily, the inverse procedure can be carried out as well, in which one starts with a theory that contains gauge-like fields on the links (without gauge symmetry), and unitarily couples them minimally to matter. It is another manifestation of the fact that the Gauss law is explicitly solved for the matter but cannot be solved for the gauge fields: as argued in the beginning, introducing gauge fields to globally invariant matter theories by minimal coupling cannot be done unitarily, but introducing matter fields that will be minimally coupled to vector fields is possible unitarily, using the inverse of the above procedure.



Figure 3: Pictorial representation of a matrix product state. The corresponding mathematical description is detailed in (134). Physical indices/legs are drawn as wavy lines while virtual indices/legs are straight.

# Acknowledgments

The first version of these lecture notes was prepared for two lectures given by Erez Zohar in the Focus week *Tensor Networks and Entanglement* of the workshop *Entanglement in Quantum System*, at the Galileo Galilei Institute for Theoretical Physics (GGI), Florence, Italy in June 2018. The current version was prepared by both authors, including further information and details.

Patrick Emonts acknowledges support from the International Max-Planck Research School for Quantum Science and Technology (IMPRS-QST) as well as support by the EU-QUANTERA project QTFLAG (BMBF grant No. 13N14780).

# A    Tensor Network Notation

This short introduction to tensor network notation is added in order to make these lecture notes self-contained and does replace not further reading [10, 11, 32]. For the ease of description, we will have a look at an arbitrary state of a spin system and rewrite it as a matrix product state (MPS). This description can be readily generalized to the description of two-dimensional systems with a PEPS description.

We consider an arbitrary state of a spin system of $N$ spins – say spin-$\frac{1}{2}$ and write it as a superposition of states in the z-basis

$$|\Psi\rangle = \sum_{\sigma_1 \ldots \sigma_N} c_{\sigma_1, \sigma_2 \ldots \sigma_N} |\sigma_1 \sigma_2 \ldots \sigma_N\rangle. \tag{133}$$

In general, the coefficients $c$ depend on the configuration of all spins in the system. As an Ansatz, we can rewrite the coefficients $c_{\sigma_1, \sigma_2 \ldots \sigma_N}$ as a product of matrices (hence the name, MPS)

$$|\Psi\rangle = \sum_{\sigma_1 \ldots \sigma_N} \underbrace{\sum_{a_1, \ldots, a_{N-1}} A_{1, a_1}^{\sigma_1} A_{a_1, a_2}^{\sigma_2} \cdots A_{a_{N-2}, a_{N-1}}^{\sigma_{N-1}} A_{a_{N-1}, 1}^{\sigma_N}}_{c_{\sigma_1, \sigma_2 \ldots \sigma_N}} |\sigma_1 \sigma_2 \ldots \sigma_N\rangle. \tag{134}$$

The objects $A$ are called *tensors* and constitute the elementary building block of the Ansatz. We call them tensors since they are objects with more than two indices. We do not imply any transformation properties generally associated with the name. They carry two types of indices: *physical indices* $\sigma$ and *virtual indices* $a$. While the physical ones correspond to the indices of the coefficients in (133), the virtual indices have no physical meaning. They are added in order to define the contraction of the tensors.

Equation (134) is depicted in Figure 3. The squares correspond to the tensors $A$ and have three legs which correspond to the three indices. As a convention for these notes, we draw the physical indices $\sigma$ with wavy lines and the virtual legs as straight lines. If a leg connects two tensors, it will be contracted, i.e. the summation of this index will be executed. Since all virtual indices in the formulation of (134) are summed over, they are all connected.

If the system of interest has more than one spatial dimension, we can adapt our description by adding more virtual indices and describe the system with a two-dimensional PEPS. The pictorial representation of PEPS is shown in Figure 4. As for the MPS case, virtual indices are depicted as straight legs and indices are contracted if their corresponding legs are connected.

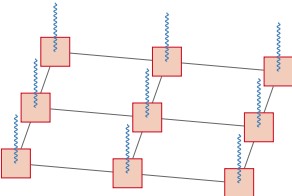

Figure 4: Pictorial representation of a PEPS. The virtual degrees of freedom connect tensors in the x- and y-direction since the system is three-dimensional. Physical indices/legs are drawn as wavy lines while virtual indices/legs are straight.

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
