# Peer review of "Gauss Law, Minimal Coupling and Fermionic PEPS for Lattice Gauge Theories"

_SciPost_

## Round 2 · Referee Report · Luca Tagliacozzo (Referee 1) · 2019-11-20

Strengths

1- pedagogical review 2- short enough 3- timely 4-original approach

Weaknesses

1- lack of historical perspective with citation to relevant papers 2- too strong statement that are not supported by equivalent theorems

Report

Overall I enjoyed reading these lecture notes, from my point of view they are a good addition to the available literature on the subject. I however think that the introduction should be expanded slightly in order to give a better account of the related works on the subject.

The justification on how one should gauge a tensor network state, or an Hamiltonian is introduced gently by starting from gauging a single system of two fermions, a nice pedagogical choice. They then proceed by showing that the idea of gauging a system by acting locally with unitary transformations on a globally invariant matter system results too restrictive.
Unless one is happy with extending the system including auxiliary matter degrees of freedom.

Here there are multiple way to do this, and one of the natural choices result into a tensor network state, the gauge invariant FPEPS. In this way tensor networks are presented as the natural tool for gauging the "extended" matter system locally.

At the end of the review however they show that the ideas do not need to rely on tensor networks, generalizing what has been done for bosons in the Fradkin and Shenker paper in the 80s. It is indeed possible to obtain a lattice gauge theory by acting on matter states with local unitaries.

The authors decide to present the opposite process, in section 4.3 where they show how to generalize the known duality that allows disentangling matter fields from gauge fields locally in order to disentangle the fermions from the gauge degrees of freedom via a local unitary transformations.

This seems more interesting since it allows to study fermionic gauge theories without fermions (though it is not at all clear that the resulting bosonic theory is simpler to characterize than the original one containing the fermions).

However the process of entangling fermions with gauge bosons could have been presented already (or at least meantioned) at the end of section 2.3, thus showing that a slight modification of what is attempted there (entangling gauge bosons with fermionic matter with local unitary transformations) is possible even without using tensor networks.

In the present form of the review, this message is some how hidden and it is easy to get the wrong picture that there is actually no way to locally entangle two different systems (one containing the matter and the other the gauge bosons) resulting in a usual gauge theory using local unitary transformation.

The authors try to clarify this point a the end of section 2.3 in the sentence about "modifying" rather than "transforming". I believe however that it would be appropriate to expand that sentence into a full paragraph and better explain the physical and conceptual consequences. As it is now, it sounds very cryptic and what modifying and transforming mean is not clear at all.

On a separate ground I suggest that the introduction should be extended.
It would be important to have an introduction that gives a better perspective of the field.

For example the use of tensor networks as a natural tool to force the the Gauss Law and extract the physical Hilbert space of a Gauge theory embedded into a larger tensor product Hilbert space has been originally discussed without relation (or with an implicit relation) to the problem of minimally coupling the matter to gauge fields.

In the high energy community, the identification of the gauge invariant Hilbert space has been done by finding dualities between gauge invariant systems and spin systems. A nice review on the subject is the one by Robert Savit Rev. Mod. Phys. 52, 453 (1980) that I guess deserves to be mentioned.

The necessity to embed the Hilbert space of gauge theories into a tensor product structure containing also to non-gauge invariant configurations has also been discussed in the context of lattice gauge theories when people started to be interested in measuring the entanglement entropy. (See for example Buividovich and Polikarpov Phys.Lett.B670:141-145,2008).

From the point of view of tensor networks, the first paper that has made connection with the above results and the possibility to use tensor network in order to describe states in the gauge invariant Hilbert space is the one by Tagliacozzo and Vidal published in Phys. Rev. B 83, 115127 (2011).

A discussion along these lines would only require a couple of extra paragraphs in the introduction and would give the reader a better overview of the field.

I thus believe that a small set of modifications will improve these already nice lecture notes.

Requested changes

1- Improve the introduction by adding the citations to the work described in the report where appropriate. 2- Define Dirac Gamma matrice after (2). 3- The notation in (12) is a bit unfortunate, consider replacing j by another letter (just a suggestion). 4- After (13) in locally gauge invariant locally and gauge actually mean the same thing, chose either one or the other. 5- In section 2.3 state explicitly that there are many different ways of "fixing the gauge" via enforcing the Gauss law. Eq 41 could seem the more natural one but this is not the only one, please mention it. 6- I am a bit confused with 42 since I would have expected that the sum on the psi^dagger psi part, not on the phases. 7- Review the three last paragraphs of section 2.3 to accommodate the observations made in the Report section and better explain what is meant. 8- First line of section 2.4 specify what this refers to. 9- From the discussion in the text I do not understand why two fermionic modes per auxiliary leg are necessary. I understand this allows to avoid adding extra tensors. Is it equivalent to add an extra link tensor with two fermionic modes encoding the electric field as (72) that is then contracted with a fermionic peps tensor with just one fermionic mode per auxiliary link? 10 First line of 4.2 change this of for that of. 10- In section 4.3 possibly mention that the opposite of this idea is a good way to locally entangle some matter fields with gauge fields resulting in a usual gauge theory (in any dimension).

---

## Round 3 · Referee Report · Luca Tagliacozzo · 2019-12-13

Strengths

Already mentioned in the previous report

Weaknesses

Still point 2 of previous weakness partially present in the new version (see report)

Report

I am happy with the reviewed version.
I still however disagree with the last statement of their reply
My observation was:
"1- In section 4.3 possibly mention that the opposite of this idea is a good way to locally entangle some matter fields with gauge fields resulting in a usual gauge theory (in any dimension)."
The authors' reply is

"Here we did not make any change. As argued in the first sections of the lecture notes, disentangling gauge fields from matter is in general not possible unitarily (in more than 1+1d) because of the non-uniqueness of the Gauss law solution. For this reason, we do not see how the opposite process of entangling matter with gauge fields would be possible - unless some configurations of electric fields are arbitrarily chosen. For this reason we decided not to include this remark in the manuscript."

We obviously disagree on this point.
In order to understand if it is a lexical disagreement or a disagreement on the physics I attach here a short note where I explicitly perform the calculations I was referring to in my comment.

They show that in any dimension we can locally disentangle the matter from the gauge fields using unitary transformations.
The final Hamiltonian is just the one of bosonic matter minimally coupled to gauge fields. This means that minimal coupling can be obtained using local unitary transformation in arbitrary dimension. The opposite to what the authors claim in their reply.

Isn't the result they present in section 4 achieving the same for fermionic matter?

Maybe we are trying to say the same thing using a different language?
I would appreciate if the authors could comment on this, and why my example works while they state that it is impossible to disentangled matter from gauge fields locally

Requested changes

1- Answer my question taking into account the attached notes

Attachment

---

## Round 3 · Author Response

We would like to thank Luca Tagliacozzo for his positive and constructive review of the manuscript.
We found the comments and suggestions for modifications very insightful and have modified the paper accordingly as summarized in the list of changes.

---

## Round 3 · List of Changes

Below, we provide detailed answers to the referee's suggestions and explain the respective modifications.

1- Improve the introduction by adding the citations to the work described in the report where appropriate.
We have now added a few sentences on the topic, including the requested references in section 2.2 which addresses the structure of the Hilbert space.
Although it is not the introduction section, it is still an introductory part of the lecture notes, where we believe it fits best.

2- Define Dirac Gamma matrice after (2).
Agreed and done.

3- The notation in (12) is a bit unfortunate, consider replacing j by another letter (just a suggestion).
Agreed and done -- replaced by l.

4- After (13) in locally gauge invariant locally and gauge actually mean the same thing, chose either one or the other.
Indeed, the formulation locally gauge invariant is redundant, we modified the sentence to "[...] is gauge invariant, i.e. invariant under local transformations generated by the Gauss law operators [...]".

5- In section 2.3 state explicitly that there are many different ways of "fixing the gauge" via enforcing the Gauss law. Eq 41 could seem the more natural one but this is not the only one, please mention it.
We rephrased the passage next to the equation to clarify this issue.

6- I am a bit confused with 42 since I would have expected that the sum on the psi^dagger psi part, not on the phases.
Indeed, the sum over x was missing and we added it. Nevertheless, the phases must be summed as well in order to obtain the right transformation.
This is the manifestation of the non-locality of the Gauss law solution in the transformation.

7- Review the three last paragraphs of section 2.3 to accommodate the observations made in the Report section and better explain what is meant.
We modified those paragraphs accordingly, aiming to clarify the difference between a unitary transformation and the process of minimal coupling.

8- First line of section 2.4 specify what this refers to.
We substituted this by "unitary gauging of separate building blocks".

9- From the discussion in the text I do not understand why two fermionic modes per auxiliary leg are necessary. I understand this allows to avoid adding extra tensors. Is it equivalent to add an extra link tensor with two fermionic modes encoding the electric field as (72) that is then contracted with a fermionic peps tensor with just one fermionic mode per auxiliary link?
The construction with two fermionic modes per auxiliary leg is not necessary, but rather used as an example.
We made it clear in the text.

10- First line of 4.2 change this of for that of.
Done.

11- In section 4.3 possibly mention that the opposite of this idea is a good way to locally entangle some matter fields with gauge fields resulting in a usual gauge theory (in any dimension).
Here we did not make any change. As argued in the first sections of the lecture notes, disentangling gauge fields from matter is in general not possible unitarily (in more than 1+1d) because of the non-uniqueness of the Gauss law solution. For this reason, we do not see how the opposite process of entangling matter with gauge fields would be possible - unless some configurations of electric fields are arbitrarily chosen. For this reason we decided not to include this remark in the manuscript.

Resubmission 1807.01294v4 on 20 December 2019
Resubmission 1807.01294v3 on 3 December 2019
Submission 1807.01294v2 on 8 August 2019

---

## Round 4 · Author Response

We would like to thank Luca Tagliacozzo for the second review and for taking the time to prepare the accompanying note. Indeed we agree with him on the physics, and the only disagreement was lexical. Therefore, we modified the manuscript accordingly.

---

## Round 4 · List of Changes

We modified the last paragraph of 4.3 to contain the extra information requested by the referee.

---

## Editorial Decision

unknown